# Leveraging soil diversity to mitigate hydrological extremes with nature-based solutions in productive catchments : an application and insights into the way forward

Benjamin Guillaume[1*], Adrien Michez[1*], Aurore Degré[1]

[1] Uliège – Gembloux Agro-Bio Tech, TERRA Teaching and Research Centre, Passage des Déportés 2, 5030 Gembloux, Belgium

* These authors contributed equally to this work

*Correspondence to*: Benjamin Guillaume (Benjamin.guillaume@uliege.be) and Adrien Michez (adrien.michez@uliege.be)

**Abstract.** Nature-based solutions (NbS) are increasingly being explored as effective strategies for mitigating hydrological extremes, such as floods and agricultural droughts. Among these, soil-vegetation-based approaches may play a key role in improving soil health, enhancing ecosystem services, and restoring the natural hydrological cycle in productive agricultural and forestry catchments, making these landscapes more resilient to climate change. However, the influence of local factors, such as soil characteristics, on the effectiveness of these interventions is often overlooked. This is largely due to the fact that commonly used lumped empirical/conceptual modelling approaches often oversimplify the complex interactions across soil – water processes at multiple scales. This study presents an innovative approach to modelling the effects of NbS landscape planning scenarios, explicitly simulating soil water fluxes. This approach allows for the investigation of how the spatial variability of soil properties influences NbS effectiveness in mitigating both floods and agricultural droughts. A fully distributed, physically based hydrological model was used to represent NbS at the catchment scale, simulating water fluxes (e.g., infiltration, evapotranspiration, runoff, soil water content) on a variable-resolution grid. This model relies on measurable local parameters (e.g., topography, soil properties, vegetation) to capture small-scale hydrological processes, enabling the representation of NbS scenarios through spatial and temporal parameter adjustments. Hydrological simulations were conducted for two catchments: one agricultural and one forest dominated, each integrating a specific landscape planning scenario. In the agricultural catchment, the scenario involved the integration of hedgerows, reduced tillage practices and soil pitting for maize crops. In the forested catchment, NbS focused on forest diversification, practices aimed at limiting soil compaction in forest and restoration of peatlands. NbS effectiveness was assessed not only downstream but also where they are implemented within the catchment using key spatial indicators. Among them, an indicator evaluating the susceptibility of vegetation cover to water stress at a given location was newly developed. The modelling approach was validated by accurately reproducing river discharge and saturated zone dynamics. It effectively captures soil natural drainage characteristics and provides a reasonable representation of NbS effectiveness, as indicated by consistency between simulated and literature values. A key finding of this study is that NbS effectiveness in enhancing flood and drought resilience strongly depends on soil natural drainage characteristics. In well drained soils, hedgerows significantly enhanced infiltration by

improving soil hydraulic properties and creating additional air space in the soil's porosity through higher rates of evapotranspiration. In contrast, improving hydraulic properties in waterlogged soils had minimal impact on infiltration due to saturation, with anoxic conditions potentially limiting transpiration. Additionally, the study highlights that well-drained soils offer co-benefits for resilience to agricultural droughts, as they are more likely to experience water deficits that NbS can mitigate. In contrast, such benefits are generally absent in waterlogged soils, which rarely face water scarcity. Results highlight that the evaluation of NbS effectiveness should recognize the spatial variability in their performance. This variability should guide the type and location of NbS to increase their overall effectiveness. The study underscores the need to move away from siloed evaluations of NbS and instead adopt a coherent, territory-based approach. It may serve as a basis for discussion and action, supporting decision-makers in implementing measures to enhance hydrological resilience. In doing so, it also reveals current knowledge gaps and identifies key avenues for future research to refine NbS effectiveness assessments, such as strengthening the availability of spatially distributed data and advancing uncertainty analysis.

## 1 Introduction

Central Europe faced consecutive abnormally dry and hot summers in 2018, 2019, and 2020 (van der Wiel et al., 2023). The drought event of 2018–2019 was considered unprecedented in the last 250 years (Hari et al., 2020), adversely impacting agricultural productivity (Toreti et al., 2019). Droughts are classified as meteorological, agricultural, hydrological, or socioeconomic. Meteorological drought refers to a deficiency in rainfall, while agricultural drought, driven by soil moisture deficits, occurs when soil water availability does not meet plant requirements. It is therefore influenced by vegetation characteristics, as species differ in their sensitivity to the same soil moisture deficit. Extended rainfall shortages may result in hydrological drought, affecting streamflow and aquifers, eventually causing socioeconomic drought when water supply fails to meet demand (Mishra and Singh, 2010).

Conversely, the same region, comprising western Germany, eastern Belgium, Luxembourg, and the Netherlands, experienced a very wet 2021 summer and an extreme precipitation event between July 13 and 16, with unprecedented accumulations. Combined with already nearly saturated soil conditions, this extreme rainfall event led to major floods, making it one of the most severe natural catastrophes in Europe of the past half-century (Journée et al., 2023).

Due to the anthropogenic global warming and the demographic expansion, the likelihood of occurrence and impact of droughts and floods is expected to be exacerbated (Aalbers et al., 2023; Dottori et al., 2023; Hari et al., 2020), consequently increasing the associated damage costs (Naumann et al., 2021), currently estimated at €9 billion for droughts (Cammalleri et al., 2020) and €7.6 billion for floods (Dottori et al., 2023) annually in the European Union.

In order to mitigate and potentially reverse these adverse effects, investments in a transformative adaptation of human systems are required (Pörtner et al., 2022). Given the uncertainty of future climate conditions, these investments could prioritize soft strategies, "no regret" approaches which yield benefits irrespective of climate change, as well as reversible and flexible options, providing significant social, economic and environmental benefits (Hallegatte, 2009). In line with this

philosophy, the recently introduced umbrella concept of nature-based solutions (NbS) is becoming increasingly popular among funders, researchers, policy makers and practitioners (Nesshöver et al., 2017). NbS could be referred to as solutions rooted in natural processes and ecosystems, designed to address a spectrum of societal and environmental challenges, including hydro-meteorological risks reduction (Ruangpan et al., 2020). NbS is a relatively broad concept that can be further specified through more targeted concepts, depending on the goals addressed, the type of ecosystem considered, and the scale of intervention. For instance, Natural Water Retention Measures (NWRM) and Natural/Small Water Retention Measures (NSWRM) refer to NbS specifically aimed at water management. Although all these concepts overlap considerably and their diverse terminology can be confusing, Magnier et al. (2024) offer a structured, targeted analysis that clarifies distinctions between them.

While NbS literature is abundant on runoff and flood risk reduction in urban areas (Ruangpan et al., 2020), there has been limited exploration of NbS as drought mitigation strategies (Yimer et al., 2024). Furthermore, potential combined effects, such as synergies and trade-offs of NbS on floods and droughts, remain largely unexplored (Fennell et al., 2023b; Penning et al., 2023). NbS are frequently considered effective in addressing hydrological droughts (Fennell et al., 2023a), and less so for agricultural droughts. However, NbS are increasingly recognized as a useful concept extending beyond the urban/riverine context, such as in agricultural or forest settings (Hanson et al., 2020) to improve drought resilience of these ecosystems while contributing to reduce flood risks. In these contexts, NbS may include soil-vegetation solutions aimed at enhancing soil health, functions and ecosystem services with agronomic and/or forest management practices and/or land restoration (Keesstra et al., 2018). However, in these alternative land use contexts such as agriculture or forestry, the effectiveness of NbS might highly depend on local characteristics such as the infiltration capacity of the soil (Fennell et al., 2023a; Penning et al., 2023). Indeed, soil hydraulic properties and their spatial distribution play a crucial role in controlling small-scale (soil profile scale) hydrological processes within a watershed (Vereecken et al., 2022) and consequently, the small-scale effectiveness of NbS influencing hydrological processes within a watershed.

This fine-scale spatial variability raises important questions about the appropriate methods and scales to assess and monitor the effectiveness of these interventions before and after implementation. One popular approach for assessing the potential effectiveness of NbS before implementation for floods and droughts involves modelling hydrological, hydraulic, and water balance processes. This approach often involves comparing the modelled outcomes of a baseline scenario of the current conditions, to a target scenario (Possantti and Marques, 2022). Several hydrological models have been used to simulate the effectiveness of NbS, including, among others, SWAT+ (European Commission, 2020), LISFLOOD (Institute for Environment and Sustainability : Joint Research Centre et al., 2012), HEC-HMS (Guido et al., 2023), or MIKE SHE (Fennell et al., 2023b). These models generally integrate different modules to calculate different water fluxes (infiltration, runoff, evapotranspiration) and can be based on empirical, conceptual, or physically-based approaches. For example, SWAT and HEC-HMS use the well-known Soil Conservation Service (SCS) method to calculate water infiltration into the soil based on an empirical formula and the soil curve number (CN). Other models, such as MIKE SHE, are process-based and use physical equations, such as Richards equation, to model infiltration. Some models allow for fully distributed simulations, while others

are lumped models or adopt a non-distributed approach. Each model has its own advantages and limitations, and the choice depends on the specific needs of the study in terms of modelling capabilities, as well as the required spatial and temporal scale and resolution, and the available data (Kumar et al., 2021b). In the urban context, the hydrological effectiveness of NbS is often assessed downstream through simulated discharge series using empirical or conceptual (sometimes lumped) rainfall runoff models (Possantti and Marques, 2022; Kumar et al., 2021). However, this type of evaluation does not differentiate the effectiveness of individual upstream interventions, ignoring the significance of the spatial arrangement of NbS within the watershed. This oversight presupposes that it has a minimal impact on their effectiveness downstream, as identified in the case of green roofs (Qiu et al., 2021).

In other land uses such as agricultural, this consideration might not be true and might require spatially distributed, physically based hydrological modelling approaches, able to rank most effective NbS-locations combinations in a given context (Brauman et al., 2022). Indeed, these types of models enable the virtual implementation of a set of NbS at the catchment scale with a specific spatial arrangement. Water fluxes, calculated on a variable-resolution grid, are derived from physically meaningful equations that rely on physically meaningful local parameters that can be measured (topography, geology, soil hydraulic properties, surface roughness, vegetation). These factors influence small-scale (grid cell scale) hydrological processes, such as infiltration. Through the spatial and temporal adjustment of these parameters, scenarios of NbS can be represented. Thus, the physical response (and effectiveness) of an intervention implemented at a specific location that might affect the global catchment's behaviour can be modelled.

This article introduces a reproducible methodology for developing a physically based surface hydrological model designed to evaluate the impact of NbS scenarios at the catchment scale. Serving as a foundational step, this modelling framework facilitates a series of innovative post-processing analyses to assess NbS hydrological effectiveness. The paper addresses four key objectives: i) Building a modelling approach that represents both the spatial variation of soil characteristics (focusing on soil natural drainage) and spatialized NbS scenarios, as well as their interactions ii) Moving beyond a simple analysis of outflow discharges by emphasizing the influence of spatial variability in soil characteristics on the effectiveness of hydrological NbS measures inside the watershed; iii) Evaluating the impact of NbS on different hydrological extremes, specifically floods and agricultural droughts; iv) Investigating synergies and trade-offs between floods and agricultural droughts mitigation through NbS implementation.

## 2 Materials and Methods

### 2.1 Study area

The Vesdre catchment was selected as the study area because it was deeply impacted during the 2021 mid-July flood event in Belgium. More than 20 people lost their lives in this valley (Dewals et al., 2021), and many residents lost some or all of their belongings. This tragic event remains deeply imprinted in the memory of the region. Unfortunately, climate projections indicate that such extreme events could occur two to three times by 2050 in this area. At the same time, these projections

suggest that summers will become increasingly hotter and drier on average (Fettweis, 2024). This situation is forcing local authorities to rethink the valley's future to enhance its resilience to the growing frequency of extreme events, including floods and droughts. A series of studies have been conducted, including the "Schéma Stratégique Vesdre" (Inondations - Reconstruction, 2024), which establishes a vision of resilience and solidarity for the catchment. It proposes a coherent set of actions focused on adaptation to climate change, addressing both floods and droughts. In this study, these actions, tailored to

the predominant land uses in the region, are tested through hydrological simulations of selected sub-catchments.

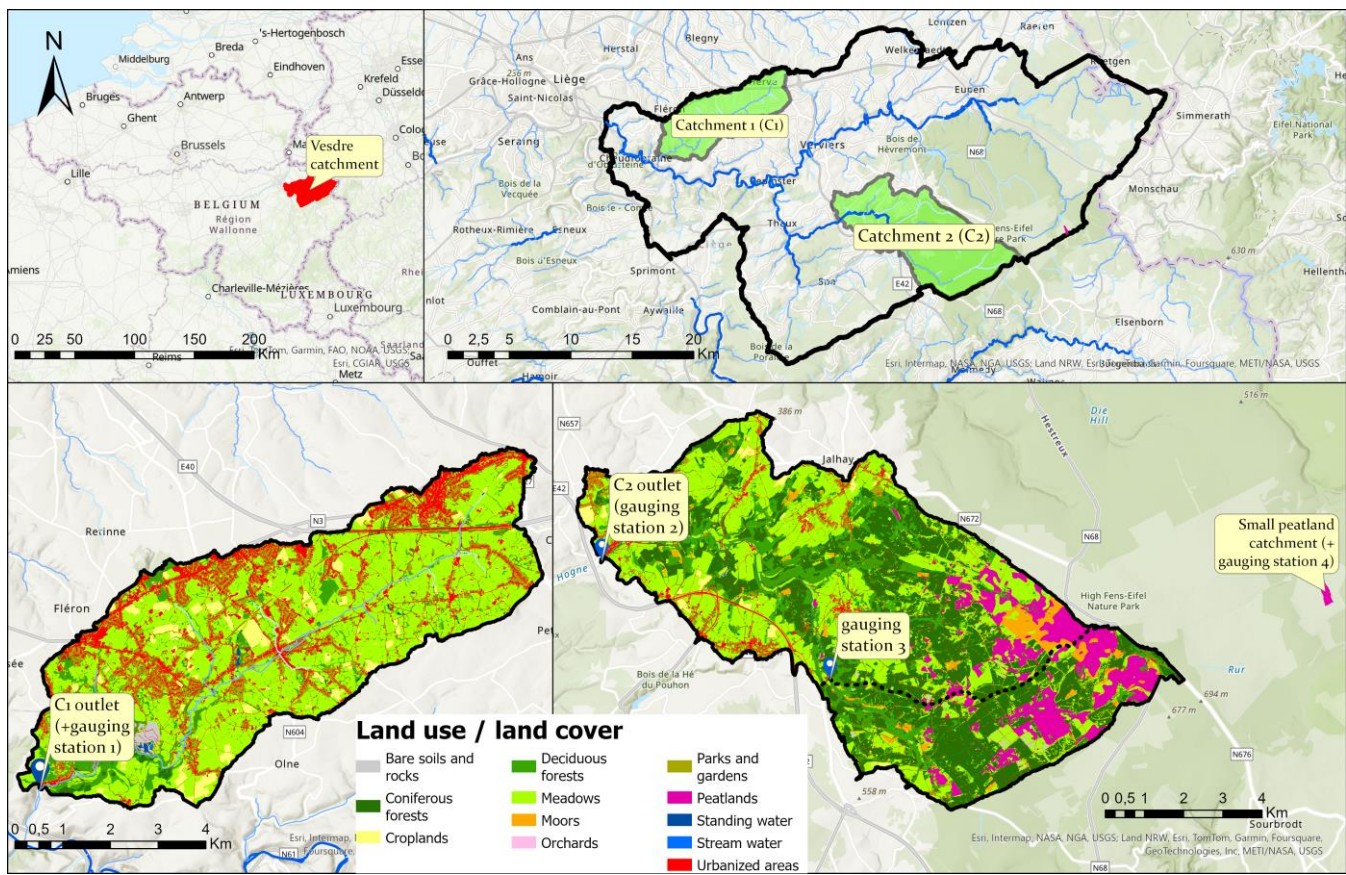

**Figure 1: Study area with maps of land use/ land cover.**

Two sub-catchments with contrasting land uses belonging to the Vesdre catchment in Belgium were selected (Figure 1). The first catchment (C1) is primarily agricultural (69 %) and peri-urban (17 %), covering 39.9 km² with altitudes ranging from

125 m to 325 m. The climate is temperate oceanic with annual precipitation rate over the catchment of 952 ± 96 mm (mean and standard deviation calculated between 2004 and 2021).  The second catchment (C2) is dominated by forests (45 %), meadows (35 %) and peatlands (10 %). It spans 71 km² with elevations ranging from 350 m to 687 m. The terrain varies from relatively flat upstream on the High Fens Plateau (dominated by peatlands, wet moorlands and drained plantations of coniferous trees) to deep valleys downstream (dominated by coniferous plantations, deciduous forests and meadows). Since

the latter half of the 19th century, extensive surface drainage efforts have been undertaken on the upstream plateau to facilitate the establishment of conifer plantations (mainly Picea abies). This has resulted in the degradation of the region's open wetlands and peat soils. The annual precipitation rate over the catchment is 1158 ± 123 mm (mean and standard deviation calculated between 2004 and 2021).

### 2.1.1 Soil and geological setting

The Vesdre catchment is characterized by a complex geological history, spanning from the Cambrian to the Holocene, marked by two orogenic cycles (Caledonian and Variscan). This evolution has resulted in significant lithological and pedological diversity. The nature of the soils in the studied catchments is shaped by two main contributions.

A large portion of the soils is developed on loess deposited during various glaciation phases, particularly the Würm. These loess deposits, rich in silt, have covered the entire region, especially on plateaus and areas with moderate relief. On plateaus

and gentle slopes, loess is often preserved, forming a homogeneous cover that promotes good permeability and, consequently, effective drainage. These silts, relatively uniform and free of coarse fragments, are still found today in catchment 1 (C1), where they generally reach a thickness of 1 to 2 meters. In contrast, in catchment 2 (C2), these deposits persist only in isolated areas and are generally associated with weathering residues from the underlying bedrock.

On steeper slopes, erosion tends to displace these deposits, reducing their thickness or even removing them completely in

certain locations. In valley bottoms, colluvium and alluvium represent the parent material for most soils.

In parallel, the degradation and weathering of the underlying rocks contribute to soil formation. The resulting fragments mix with loess deposits and often reflect the lithological nature of the bedrock. The intensity of this process varies depending on topography and rock resistance. In areas where loess is preserved, such as plateaus and moderate slopes, the contribution of rock fragments tends to be less significant, resulting in soils that are less stony. Conversely, on steep slopes and in areas

where loess has been largely eroded, the influence of the underlying rock becomes more pronounced, leading to soils with a higher coarse fraction and smaller depths of loose material.

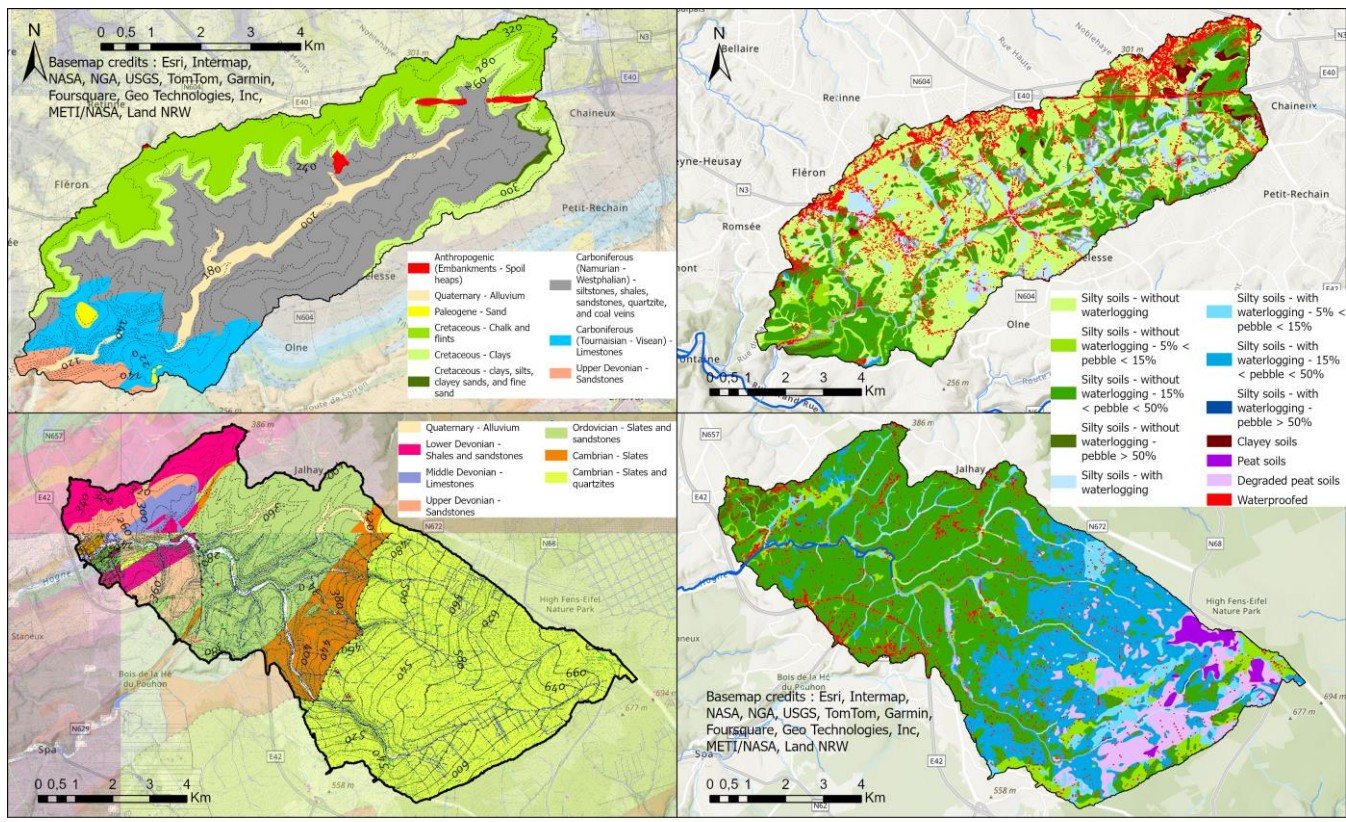

**Figure 2 : (Geological maps of C1 and C2, showing the absolute age of geological units (Period, Epoch, or Age) and the principal rock types within these units (Carte Géologique de Wallonie, 2025; Carte hydrogéologique de Wallonie, 2025). Soil maps of C1 and C2, depicting textural classes, natural drainage characteristics, and stoniness (Carte Numérique des Sols de Wallonie, 2023).<**


Soil natural drainage primarily depends on several factors. The permeability of the substrate and the nature of the materials play a crucial role. Soils derived from recent deposits, such as colluvium, alluvium, and loess, are generally permeable, facilitating infiltration. However, soils formed from more altered or clay-rich units, such as the Cretaceous Herve smectites (locally present in C1), exhibit reduced permeability, leading to slower drainage.

The presence of impermeable layers also influences drainage. In certain areas, particularly upstream of C2, alteration clays or solifluction layers act as barriers, restricting percolation and leading to the formation of perched water tables. Under these conditions, the decomposition of fresh organic litter slows significantly, resulting in a peat-forming soil. If this process continues, it can eventually lead to the formation of sphagnum peat, as observed in the extreme Eastern part of C2. Conversely, when soils lie directly on a more permeable substrate, drainage is improved. This is notably the case in the lower part of C2, where solifluction layers tend to be thinner and soils directly overlying schists benefit from the rock's schistosity, which enhances infiltration. Similarly, in the lower part of C1, limestone formations may also contribute to efficient drainage, despite the fact that the loess overlying these limestones tends to be more clay-rich.

Topography further influences drainage conditions. In valley bottoms, the water table is often permanent and reaches the surface, limiting infiltration in these soils. Locally, peat soils can be found in valley bottoms of C2. In contrast, on plateaus and gentle slopes covered with loess, such as in C1, drainage is generally good, except in certain areas of the upper part of C2, where clay accumulations or the presence of impermeable layers may restrict infiltration.

## 2.2 Modelling Framework

### 2.2.1 Hydrological model

The model selection was based on its capability to represent long time series of multiple hydrological processes (channel flow, overland flow, unsaturated- and saturated-flow, evapotranspiration, etc), accouting for interactions between saturated zone, vadose and surface, at the catchment scale using a fully distributed 3D lattice. Therefore, the coupled hydrologic/hydraulic MIKE SHE/MIKE 1D model was chosen. MIKE SHE/MIKE 1D solves partial differential physically based equations describing mass and energy transfers. The actual evapotranspiration is derived from the reference evapotranspiration using the Kristensen and Jensen method. 2D overland flow is calculated using the diffusive wave approximation of the Saint-Venant equations. 1D Channel flow is derived from the fully dynamic higher order approximation of the Saint-Venant equations. 1D Unsaturated flow is derived from the Richards equation. Saturated flow is described by the 3D Darcy's equation numerically solved using the finite difference method (DHI, 2024). All input data are summarized in Table 1.

**Table 1 : Summary of model input data.**

| Data type | Data source | Resolution | | Description |
| | | Spatial | Temporal | |
|---|---|---|---|---|
| Rainfall | Gridded observational data | 5 km | Daily | Gridded and point observations are combined to obtain hourly 5 km gridded rainfall |
| | https://hydrometrie.wallonie.be | Point observations | Hourly | |
| Reference evapotranspiration | Gridded observational data | 5 km | Daily | Estimated with the Penman-Monteith equation (Allan et al., 1998) |
| River discharge | https://hydrometrie.wallonie.be | Point observations | Hourly | To calibrate and validate the models |
| DEM | LIDAXES (version 2) - MNT, 2023 | 2 m | 2013-2014 | Hydrologically conditioned DEM |
| LULC | WALOUS 2018 - Série, 2024; Bassine et al., 2020 | 5 m | 2018 | This data served as model input to describe the spatial distribution of LAI, Kc, Rd and M value of Manning. |
| Soil | (Carte Numérique des Sols de Wallonie, 2023) | 1/20.000 | 1947-1991 | The soil map of Belgium was used as input to identify homogeneous soil unit, stone content and soil depth. It is also used to validate the model with respect to the natural drainage classification. |
| | (Textures et fractions granulométriques de référence des sols de Wallonie - Série, 2023) | 50 m 3 depths (0-40 cm, 40-80 cm, 80-120 cm) | 1947-2020 | Map of textural fractions: clay (0–2 μm), silt (2–50 μm), and sand (50–2000 μm). To each layer of each homogeneous soil unit is assigned mean values of textural fractions. |
| | Updated European hydraulic pedotransfer functions (euptfv2) (Szabó et al., 2021) | / | / | Retention and hydraulic conductivity curves each layer of each homogeneous soil unit were derived from the pedotransfer function 1 using depth and mean values of textural fractions as predictors. |
| Geology | Sohier, 2011 | 1 km | / | This data originates from a Bibliographic study of the hydrogeological context of the Vesdre catchment. It gives the spatial repartition of homogeneous hydrogeologic units and an initial guess of hydraulic conductivities, which was refined during the model calibration. |

Hydrometeorological data: The climate data comprised a 5 x 5 km grid detailing hourly precipitation intensities and daily reference evapotranspiration. Reference evapotranspiration was estimated using the Penman-Monteith equation (Allan et al., 1998) applied to data collected by the observation network of the Belgian Royal Meteorological Institute (RMI), including temperature, humidity, wind speed, and solar radiation (« Gridded observational data »). Precipitations were determined by combining RMI's gridded daily data with hourly point measurements from the Wallonia Public Service's rain gauge network (https://hydrometrie.wallonie.be), enabling the hourly distribution of RMI's daily rainfall data.

Hourly stream discharge data were obtained from 11 May 2011 at the outlet of C1 (gauging station 1), from 30 November 2015 at the outlet of C2 (gauging station 2) and from 1 January 2000 in an upstream branch of C2 with (gauging station 3) a drained area of 19.6 km² (https://hydrometrie.wallonie.be).

Topographic data: The topographic data were a hydrologically corrected Digital Elevation Model (DEM) at a 2m resolution (LIDAXES (version 2) - MNT, 2023). Resampling onto the model grid involves selecting the minimum elevation value to maintain the hydrological correction applied.

Land cover – land use data: Land use/land cover (LULC) information was used to define spatially variable surface roughness and vegetations. The LULC map was a 5 m resolution layer of the land cover of 2018 (WALOUS 2018 - Série, 2024; Bassine et al., 2020) mapping several LULC classes. Resampling onto the model grid was performed by selecting the predominant LULC within each grid cell. Each LULC class was assigned an M value of Manning.

The surfaces were also categorized into seven vegetation classes based on the LULC map. Apart from permanent grasslands, which were considered open production surfaces, most croplands were cultivated with maize. Consequently, maize development dynamics were assigned to all croplands, and in the absence of information on cropping practices, no crop rotations were modelled. Each vegetation class was assigned annual dynamics (temporally varying) of leaf area index (LAI), crop coefficient factor (Kc), and root depth (Rd) (Figure - A1). These parameters are used by the model to refine the reference evapotranspiration (Penman-Monteith) into the actual evapotranspiration following the approach of Kristensen and Jensen (DHI, 2024). In addition, detention storage, which is a parameter aimed at accounting for ponding in depressions at sub-cell scale, was fixed to 4 mm in both catchments.

**Table 2: Equivalences between land use / land cover classes and vegetation classes.**

| Vegetation class | Land use / land cover class | Maximum rooting depth (cm) |
|---|---|---|
| No vegetation | Railway network, road network, above ground artificial construction, artificial ground covering, and bare soil | 0 |
| Wetland | Bog, lake, pond, pool, basin, wet heathland, river surface, | 10 |

| | and other unclassified water surface | |
|---|---|---|
| Open production surfaces | Herbaceous cover in rotation within the year, herbaceous cover all year round, forage crop, hay meadow, wet meadow, permanent meadow, intensive permanent meadow, orchard, nuts and other herbaceous cover all year round | 50 |
| Coniferous Forests | Spruce forest, Douglas fir forest, pine forest, conifers, conifers < 3m, conifers > 3m, Christmas tree, and other coniferous stand or unknown resinous essence > 3m | 60 |
| Deciduous Forests | Deciduous trees, deciduous trees < 3m, deciduous trees > 3m, birch forest, oak forest, beech forest, larch forest, poplar forest, other deciduous stand or unknown deciduous essence > 3m | 120 |
| Open conservation zones | Dry heathland and dry grassland | 50 |
| Croplands | Corn, cereal and similar, beetroot, chicory, potato, vegetable, oilseed and other crops or other agricultural uses | 40 |

Soil and subsoil data: The 1D Richards equation was used to model vertical water fluxes in the unsaturated zone, which required the determination of the soil hydrodynamic properties (soil water retention and soil hydraulic conductivity curves) using the Van Genuchten and Mualem functions (Van Genuchten, 1980) at each location within the unsaturated zone. Saturated flow was modelled with the 3D Darcy's equation requiring vertical and horizontal saturated conductivities to be specified at each location within the saturated zone.

The soil and subsoil were represented as a series of layers with homogeneous hydrodynamic properties within each layer, but varying between layers in terms of thickness and lateral arrangement. Except for peat and degraded peat soils, the upper soil profile was discretized into two layers: topsoil, ranging from 0 m to 0.4 m, and subsoil, extending from 0.4 m to a maximum of 2 m deep. These two layers could be truncated by the base at depths of 0.2 m, 0.3 m, or 0.6 m, considering the soil depth information from the Belgian soil map (Carte Numérique des Sols de Wallonie, 2023).

The upper soil profile was delineated into homogeneous units based on the Belgian soil map (Figure 1) (Carte Numérique des Sols de Wallonie, 2023). This map is derived from approximately 6 000 000 samplings conducted with a soil auger (max 1.25 m depth) between 1947 and 1991 of the Belgian cadastral plans, according to a square grid of 75 m per side (1 to 2.5 observations per hectare). This map delineates 6000 soil units defined by characteristics such as texture, natural drainage, diagnostic horizon, and stone content (Legrain et al., 2011). For each homogeneous unit, excepted for peat, degraded peat and impermeable soils, retention and hydraulic conductivity curves (Van Genuchten and Mualem models, with m = 1-1/n

and L = 0.5 ) for both soil layers were derived from the European hydraulic pedotransfer function (euptfv2) number 1 using depth and the averaged soil texture (« Textures et fractions granulométriques de référence des sols de Wallonie - Série ») as predictors (Szabó et al., 2021). The predicted saturated water content, $\theta_s$, was subsequently corrected to account for the loss of porosity due to stone content, $s$, as represented in the Belgian soil map.

$$\theta_{s-pebble} = \theta_s(1-s) , \tag{1}$$

For the peat soil units, retention curves were obtained by measurements of water content on peat samples of local bogs using pressure plates at pF values of 1.00, 1.60, 1.85, 2.00, 2.30, 2.78, 2.95, 3.62, 4.14. Van Genuchten functions were then adjusted to measured retention curves. The saturated hydraulic conductivity of peat is spatially highly variable and challenging to measure in laboratory. Saturated hydraulic conductivity was initially determined based on values found in the literature (Wastiaux, 2008). As suggested by Szabó et al. (2024), it was subsequently refined through calibration using discharge measurements conducted between 2012 and 2015 at the outlet (gauging station 4) of a small peatland catchment of 14 hectares, located 5 km away from C2. The thickness of the degraded peat soil unit was assumed to be 0.4 m, while the peat soil unit was assumed to be 2 m thick. Beneath peat or degraded peat, a low-permeability layer of 0.6 m thickness with 60 % clay, 35 % silt, and 5 % sand was assumed, and its hydrodynamic parameters are retrieved using euptfv2.

Below the upper soil profile, multiple geological units extending to a depth of 18 m were delineated based on a 1x1 km grid. Each geological unit consisted of multiple layers, exhibiting variations in thickness and vertical hydraulic conductivity. These values were defined through a bibliographic study of the hydrogeological context of the Vesdre catchment (Sohier, 2011). Due to their imprecise nature and large spatial variability, vertical hydraulic conductivities were initially defined as ranges of plausible values and were subsequently refined during the model calibration process. Horizontal hydraulic conductivities were assumed to be 10 times greater than vertical hydraulic conductivities (DHI, 2024).

Models' specification, calibration and validation: For both catchments, a model of the present situation (further referred to as BASELINE) was calibrated and validated. The total simulation period was comprised between 1 January 2002 and 31 December 2021. At the start of the simulation process, the soil water content was set to field capacity. The first two years of simulation were considered a warming period and were not considered in the data analysis. Outputs were stored on an hourly basis. To ensure a fine representation of the catchments while maintaining reasonable computational times, the grid resolution of the C1 model was 20 m, whereas it was 40 m for the C2 model. Calibration was conducted in the period between 1 October 2017 and 1 October 2021, while validation encompassed the remaining available data.

The calibration was performed manually and limited to very few parameters, for which the available data were considered uncertain and had the most impact (the most sensitive parameters) on the modelled hydrographs. These parameters included the Manning's M roughness coefficients for overland and channel flows (which are mostly sensitive regarding the temporality of hydrographs and peak flows), as well as the saturated hydraulic conductivity of the geological layers (which are sensitive regarding baseflow, peak discharge values, and groundwater head) (XEVI et al., 1997). Calibration of the

Manning's M roughness and the saturated hydraulic conductivities of peat were performed using discharge measurements conducted between 2012 and 2015 at the outlet of the small peatland catchment of 14 hectares (gauging station 4). Calibration and validation were performed in relation to historical discharge measurements taken at the river gauging stations. The ability of the model to reproduce observed discharges was assessed against the Moriasi et al. (2007) model evaluation guidelines. This included the computation of the Nash-Sutcliffe Efficiency (NSE), the ratio of the root mean square error to the standard deviation of measured data (RSR), and the percent bias (PBIAS).

$$NSE = 1 - \frac{\sum_i (QObs_i - QMod_i)^2}{\sum_i (QObs_i - \frac{\sum_i QObs_i}{n})^2},\tag{2}$$

NSE varies from -∞ to 1. The closer to 1 the better the simulation performance. $QObs$ and $QMod$ are the observed and simulated discharges, respectively. $i$ is the time step.

$$RSR = \frac{RMSE}{STDEV_{obs}} = \frac{\sqrt{\sum_i (QObs_i - QMod_i)^2}}{\sqrt{\sum_i (QObs_i - \frac{\sum_t QObs_i}{n})^2}},\tag{3}$$

RSR varies from 0 to +∞. The closer to 0 the better the simulation performance.

$$PBIAS = \frac{\sum_i (QObs_i - QMod_i) * 100}{\sum_t QObs_i},\tag{4}$$

PBIAS varies from -∞ to +∞. The closer to 0 the better the simulation performance. Positive values indicated model underestimation bias, and negative values indicated model overestimation bias.

In the absence of piezometric data at the study sites, a visual assessment of the simulated vertical dynamics of the saturated zone was conducted on soils of each drainage class of the Belgian soil map (Table 3). This ensured that the modelled saturated zone dynamics match those described by the soil map. Drainage class is a concept used in different soil classifications to describe the frequency and duration of soil wetness under natural conditions (Ditzler, 1999). In the Belgian soil classification, drainage classes are determined by the depth at which mottling and reduction processes occur in the soil profile, often corresponding to the levels of saturated zone fluctuations (Table 3). This method facilitates semi-quantitative spatial validation across the entire catchments' areas.

**Table 3: Definition of the drainage class in the Belgian soil classification for loam and clay soils.**

| | | Depth (cm) at which mottling and reduction features start | |
|---|---|---|---|
| Natural drainage class for clay and silt soils | | Mottling | Reduction |
| Without waterlogging | Favourable | >125 | - |
| | Moderate | 80-125 | - |
| | Imperfect | 50-80 | - |
| Temporarily | Quite poor | 30-50 | - |

| | Poor | 0-30 | - |
| waterlogging | Quite poor | 30-50 | >80 |
| Permanent waterlogging | Poor | 0-30 | 40-80 |
| | Very poor | - | <40 |

### 2.2.2 Landscape planning scenarios

The impact of NbS on catchment hydrology was evaluated through the comparison of responses between BASELINE and POST configurations, under consistent meteorological forcing. The POST landscape configuration represented a scenario where soil-vegetation NbS were implemented at the scale of the entire watershed. These landscape planning POST scenarios were developed in accordance with the "Schéma Stratégique Vesdre" (Inondations - Reconstruction, 2024), which outlines a long-term vision for sustainable territorial development and proposes a range of potential measures to mitigate flood and

drought risks. The modelled scenarios (POST) incorporate as many feasible measures as possible within specific contexts (agriculture-dominated and forest-dominated) to assess their potential for flood and drought mitigation. The scenarios compare the current situation to a long-term projection (2050 horizon), assuming full implementation and maturity of the measures. The transition process between current and future states is not modelled, although it may significantly influence the actual effectiveness over time. It should be emphasized that accounting for the transition phase may be essential, as some

measures require several years or even decades to achieve full efficiency, and the speed at which a measure becomes effective may serve as an important criterion for prioritizing NbS.

The following NbS measures were implemented in the agricultural catchment (C1) (Figure 3– a):

- Hedgerows: a dense network of hedgerows was systematically implemented around all agricultural parcels,
representing around 700 km of barriers within the entire watershed ($\approx$175 m. hectare$^{-1}$) (Table 4-a).
- Agricultural practices: water and soil conservation practices were modelled on cultivated land, including soil pitting for maize crops (Table 4-c) to form small depressions between rows (0.73 km² or 1.8 %) (Clement et al., 2023) and the adoption of reduced tillage practices (Table 4-b) for the other crops (0.30 km² or 0.8 %).

In the forest catchment (C2), the following NbS measures were implemented (Figure 3- b):

- Restoration of peatlands: this involved restoring peatlands and open wetlands on degraded peaty soils, currently occupied by conifer plantations and moors (4.6 km² or 6.5 %). The scenario simulated the plugging of the existing surface drainage network with 275 plugs distributed along concentrated flow paths (drainage area from 2 to 5 hectares) and the creation of 32 ponds of 640 m³ each (Table 4-f).
- Forest diversification: this scenario was complemented by forest diversification, involving the conversion of
monospecific conifer plantations into irregular mixed stands on temporarily waterlogged soils (5.4 km² or 7.6 %) (Table 4-e).
- Practices aimed at limiting soil compaction: for all forest areas, potential effect of forest practices aimed at limiting soil compaction, notably through compartmentalization with designated skid trails were incorporated into this scenario (16.8 km² or 23.7 %) (Table 4-d).

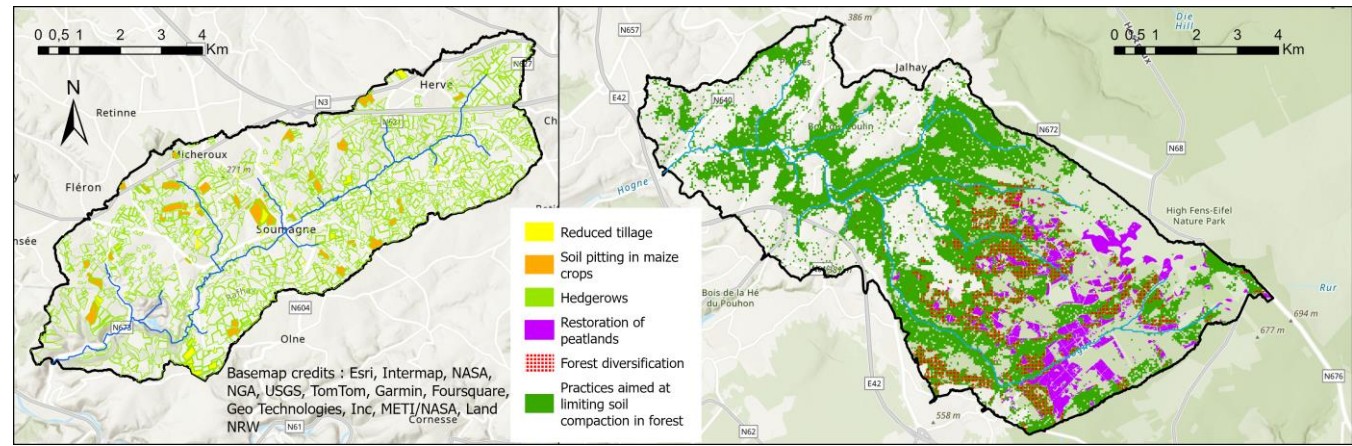


**Figure 3: Modelled NbS in POST configurations of C1 and C2.**

These NbS measures are incorporated into the model by adjusting some key parameter values (Table 4). It should be noted that although some literature exists on how NbS may influence these parameters, significant uncertainties remain regarding their exact values, as they cannot be assumed to apply directly to this specific case. These parameter values should be

considered as hypotheses, while remaining within plausible ranges.

**Table 4: Summary of hypotheses for parameter modifications in models to represent NbS scenarios.**

| Modelled NBS | Modified parameter | How modified |
|---|---|---|
| **a - Hedgerows** | Vegetation class | from BASELINE to deciduous forest |
| | Manning M | from BASELINE to 2 |
| | Saturated hydraulic conductivity of the first 40 cm of soil | 100 % |
| | Saturated water content of the first 40 cm of soil | + 20 % |
| **b - Reduced tillage practices** | Alpha shape parameter of the Van Genuchten's & Mualem's functions of the first 40 cm of soil | – 20 % |
| **c - Soil pitting for maize crops** | Detention storage | + 20 mm |
| **d - Practices aimed at limiting soil compaction in forest** | Saturated hydraulic conductivity of the first 40 cm of soil | + 50 % |
| | Saturated water content of the first 40 cm of soil | + 10 % |
| **e - forest diversification** | Vegetation class | from BASELINE to deciduous forests |
| | Manning M | from BASELINE to 5 |
| **f - Restoration of peatlands on degraded peat soils** | Vegetation class | from BASELINE to wetland |
| | Manning M | from BASELINE to 5 |
| **f - Plugs in the drainage network** | Topography | + 0.45 m |

| | | | |
|---|---|---|---|
| **f - Ponds** | Topography | - 0.45 m |

## 2.2.3 Integrated hydrological analysis of model outputs

A complete processing chain of results focusing on the quantification of hydrologic impact of NbS measures and scenarios was conducted (Table 5). Hydrologic indicators were calculated or developed to capture the impact of various measures within the watershed, emphasizing the spatial variability of their effectiveness. These indicators are computed to assess how NbS could influence hydrological extremes, particularly focusing on flooding events and agricultural droughts.

**Table 5: Summary of hydrological indicators aimed at assessing the effectiveness of NbS to mitigate floods and drought.**

| Indicator | Focus | Spatial scale of application | Temporal scale of application | Overview | Variables of interest |
|---|---|---|---|---|---|
| Peak flow percent reduction | Flood | Watershed (outlet) | Every rainfall event | Modification of peak discharges | River flow rate |
| Flood volume percent reduction | Flood | Watershed (outlet) | Every rainfall event | Modification of the total volumes discharged | River flow rate |
| Spatial variation of infiltration rates | Flood | Spatially distributed (grid cell) | Every rainfall event generating > 5 mm runoff | Modifications of spatially distributed infiltration | Infiltration, seepage flow |
| Spatial Vegetation Drought Stress Duration Frequency (sVDSDF) | Agricultural drought (all types of vegetation) | Spatially distributed (grid cell) | Days to months | Modifications of consecutive days of vegetation stress considering a stress threshold and a return period | Soil water potential, root depth |

Peak flow and flood volume percent reduction: These indicators were integrative metrics permitting to evaluate the overall effect of a given scenario on flood reduction without discriminating against the impact of the various measures that comprise the scenario. To this end, these two indicators were calculated for a set of rainfall events generating a detectable signal on the hydrograph. A focus was also made on the rainfall event between July 13 and July 18 of 2021:

- Peak flow percent reduction : The evolution of peak discharges between the BASELINE and the POST scenarios.

$$Peak\ flow\ percent\ reduction = 100 * \frac{Peak\ discharge_{POST} - Peak\ discharge_{BASELINE}}{Peak\ discharge_{BASELINE}},$$

(5)

- Flood volume percent reduction: The evolution of the total amount of water discharged at the outlet between the BASELINE and the POST scenarios.

$$Flood\ volume\ percent\ reduction = 100 * \frac{Total\ volume_{POST} - Total\ volume_{BASELINE}}{Total\ volume_{BASELINE}},$$

(6)

Spatialized indicators: These indicators aimed to assess the local hydrological impact of specific NbS measures where they were located within the watershed. This allowed for discrimination of measures based on their relative impact within the same scenario. These indicators presupposed that the hydrological effects observed at a given location are exclusively attributed to the measure implemented at that site, irrespective of the presence of other measures elsewhere within the watershed.

-    Spatial variation of infiltration rates: As a spatialized indicator to assess the effectiveness of NbS to mitigate flood, the cumulative infiltration, $I_{cum}$, between July 13 and July 18 of 2021 was calculated for each model grid cell.

$$I_{cum} = \sum_i (Infiltration_i - Seepage_i) , \tag{7}$$

    $Infiltration$ was the amount of water (in mm) flowing into the soil. $Seepage$ was the amount of water (in mm) outflowing from the soil. $i$ was the time step.


    -    Spatial Vegetation Drought Stress Duration Frequency : An indicator to assess the effect of NbS on vegetation water stress was (newly) developed, utilizing a frequency analysis. Consecutive days where plants at specific locations underwent water stress beyond a designated threshold against a return period were mapped. Thresholds were determined based on the soil water potential within the root zone. An extreme stress threshold was defined
when the water potential in the root zone drops below -150 m ($\approx$ -1.5 MPa), closely approaching the permanent wilting point which is classically recognized by soil scientists as the soil water potential beyond which most plants cannot survive (Wiecheteck et al., 2020). Additionally, a moderate stress threshold for vegetation is established at a water potential of -30 m ($\approx$ -0.3MPa), corresponding in our study sites to approximately 30% of the relative extractable water content in soil (Gourlez de la Motte et al., 2020; Granier et al., 2007). The methodology to
construct this indicator is briefly described as follows: For each cell of the model and for each year (2004-2021), the maximum consecutive duration during which the water potential in the soil profile explored by the roots (rhizosphere) was below the stress threshold was extracted. Then, for each cell, a Gumbel distribution was used to establish the relationship between the duration (y) and the probability of occurrence (return period: TM) of the stress.

$$y = -c \left( \ln \ln \frac{TM}{TM-1} \right) - a, \tag{8}$$

    c and a are adjustment parameters. TM corresponds to the return period and is calculated according to:

$$TM = \frac{n+1}{m} , \tag{9}$$

    n is the rank in the descending order and m is the total number of years of observations. Once the parameters, a and
c are fitted for a given stress threshold, y can be retrieved using eq. (8) (Maidment, 1992).

# 3 Results and Discussion

## 3.1 Calibration and validation of the baseline model

The simulated discharges evaluated at each gauging station, against Moriasi criteria, showed satisfactory to good performance (Table 6). Among the three indicators, the NSE values were the least satisfactory according to Moriasi's criteria. This could be explained by the fact that NSE is very sensitive to peak flows while measured flow rates during these events are more uncertain (well above the maximum gauged discharge on the rating curve). RSR and PBIAS were considered satisfactory/good ($0.50 <$ RSR $< 0.60$ for good, $0.60 <$ RSR $< 0.70$ for satisfactory) and very good (PBIAS $< \pm 10$).

**Table 6: Summary of model performance at gauging stations**

|  | Calibration | | | Validation | | | |
|---|---|---|---|---|---|---|---|
|  | NSE | RSR | PBIAS | NSE | RSR | PBIAIS | Model performance |
| Gauging station 1 | 0.72 | 0.53 | -7.2 | 0.64 | 0.60 | -9.3 | Satisfactory |
| Gauging station 2 | 0.69 | 0.55 | 8.0 | 0.65 | 0.59 | 1.5 | Good |
| Gauging station 3 | 0.68 | 0.56 | 14.7 | 0.62 | 0.61 | 8.6 | Satisfactory |
| Gauging station 4 | 0.74 | 0.51 | 8.6 |  |  |  |  |

The modelled dynamics of the saturated zone depth agreed well with the expected dynamics for each drainage class on the Belgian soil map (Figure 4). The range of variation, as well as the frequencies of the modelled saturated zone depths, were slightly higher in the degraded peat soils compared with the intact peat (Wastiaux, 2008). Soils designated as "without waterlogging" on the soil map were seldom, if ever, subject to waterlogging. In most grid cells, the modelled saturated zone remained well below the surface, with few exceptions in late winter. As expected in temporarily waterlogged soils, the modelled saturated zone often reached the surface during the winter months and receded during the summer. Overall, the model successfully represents the variability in the natural drainage characteristics of soils in the BASELINE scenario, thus addressing our first objective, which was a prerequisite for assessing the effectiveness of the NbS measures on various soils with differing natural drainage characteristics.

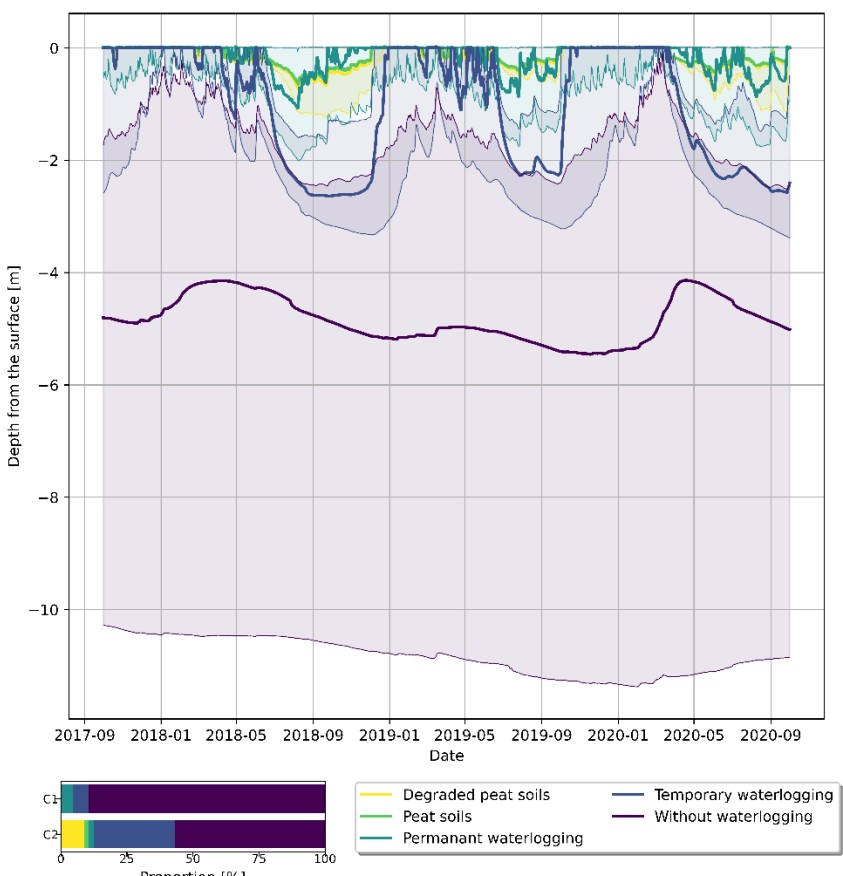

Figure 4: Modelled saturated zone depth (watersheds C1 and C2) based on drainage classes from the Belgian soil map. The thick line represents the median values calculated from each model cells for each drainage class. The thin lines indicate the 25th and 75th percentiles. The stacked bar chart represents the relative proportion of surface of natural drainage classes of soils found in C1 and C2.

## 3.2 Effectiveness of NbS scenarios against floods at the outlets

The modelled specific peak discharges under BASELINE scenario (Figure 5) indicate that, for an equal surface area, C2 generates more runoff than C1, despite being predominantly forested, a land cover generally associated with lower runoff generation compared to pasture (Bathurst et al., 2020) or urbanized areas. This result may seem counterintuitive as it does not align with the common assumption that forest-dominated catchments produce less runoff than agriculturally dominated ones. However, C1 and C2 differ in terms of topography, morphometry, precipitation rate, geology, and soil characteristics (Figure 1 and Figure 2), all of which can influence runoff production and transfer within a catchment. Figure 4 also highlights that C2 has, compared to C1, a higher proportion of waterlogged soils, especially in the upstream part (Figure 2), which hinder infiltration and promotes runoff.

The extreme rainfall event between 13 July and 18 July of 2021 was one of the events generating the highest peak discharge

(represented by the squares in the upper part of Figure 5 in the modelled time series). The modelled hydrographs of this event in both catchments are presented in the lower part of Figure 5.

Landscape planning scenarios showed contrasting modelled effectiveness in terms of reducing peak discharges between the two catchments. In C1 (mostly agricultural), modelled peak discharges were reduced by approximately 30 %, while in C2 (forest-dominated) it was approximately 10 %. Moreover, for both catchments, the effectiveness in reducing peak discharges

seemed not influenced by the magnitude of the events, as indicated by the linear trend observed in the scatter plot between BASELINE and POST peak discharges. The response remained relatively consistent for both low- and high-discharge events.

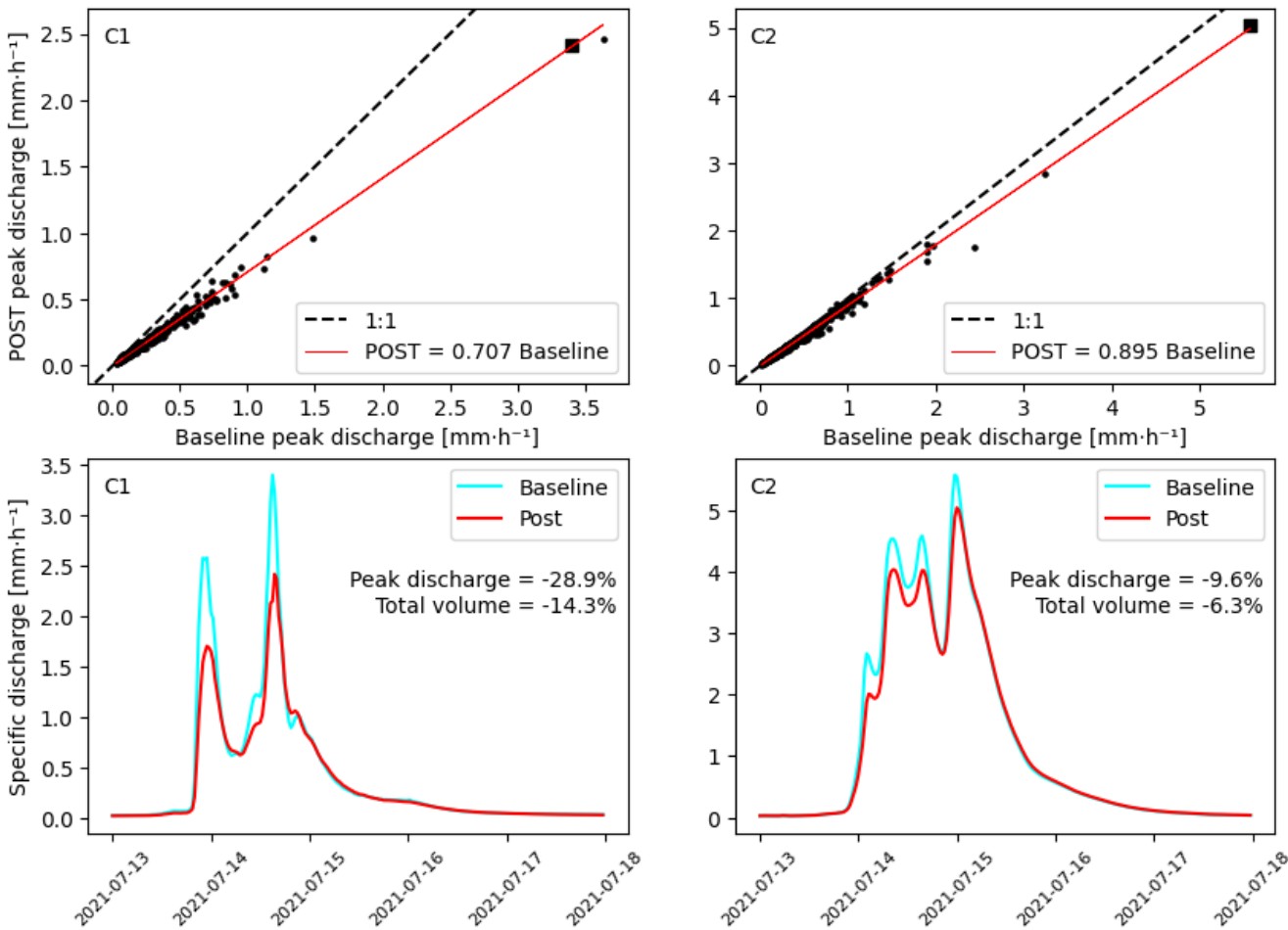

**Figure 5: Modelled specific peak discharges at the outlets of C1 and C2 before and after the implementation of landscape planning**
**scenarios. The top section shows scatter plots illustrating the evolution of peak discharges for a large number of runoff events between 2004 and 2021. The regression slope coefficients indicate the average peak flow percent reduction. The squared dot highlights the peak discharge of the event occurring between 13 July and 18 July 2021. The bottom section presents hydrographs of this specific event, comparing conditions before and after the implementation of the landscape planning scenario.**

### 3.3 Effectiveness of individual NbS in mitigating floods and agricultural droughts

The following sections explore the effectiveness of individual measures. To this end, the analysis focuses on the rainfall event that occurred between 13 July and 18 July 2021.

As highlighted in Figure 6, model outcomes indicate that NbS generally exhibit positive effects on infiltration (and runoff reduction) and the mitigation of agricultural drought. However, the magnitude of these effects seems to vary depending on the specific type of NbS and context, as observed in the previous section.

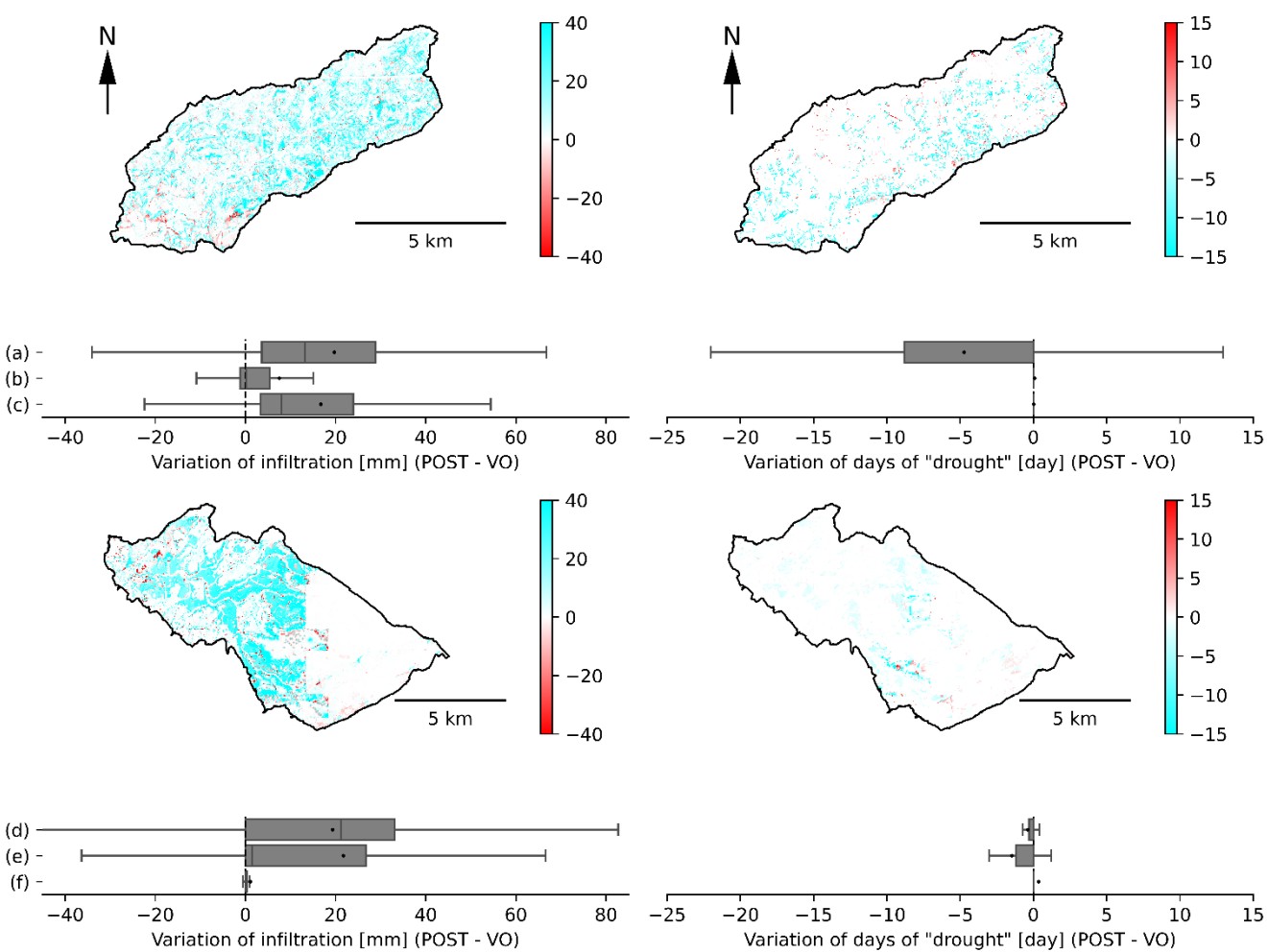


**Figure 6: Variation of cumulative infiltration (from 13 to 18 July 2021) and of Vegetation Drought Stress Duration (matric potential of -30m and return period of 25 years) on model grid cells before (BASELINE) and after (POST) implementing NbS (a : Hedgerows, b : Reduced tillage, c : Soil pitting, d : Forest practices aimed at limiting soil compaction, e : Forest diversification with practices aimed at limiting soil compaction, f : Restoration of peatlands).**

### 3.3.1 Hedgerows

In the agricultural context (Figure 6-a), model results suggest that hedgerows can be highly effective in improving infiltration during an extreme rainfall event, as reviewed by Rajbanshi et al. (2023). This modelled increase in infiltration during extreme events could be attributed to two primary factors: the enhancement of the soil's hydrodynamic properties and the generally drier soil conditions under hedgerows, which increases the unfilled soil porosity and the hydraulic gradient responsible for infiltration. Due to the lack of farm traffic, greater incorporation of organic matter, or root network promoting macropores, soils under hedgerows are often found to be less compacted and more permeable than those in adjacent arable or pasture fields (Holden et al., 2019; Wallace et al., 2021). Drier soil conditions under hedgerows could be attributed to increased actual evapotranspiration (Benhamou et al., 2013) and canopy interception. A counterintuitive observation from our simulations is that, when considering a broader temporal window, the cumulative annual modelled infiltration decreased under hedgerows due to this increased canopy interception; a larger portion of the rainfall was directly evaporated before reaching the soil for smaller and more common events (results not shown). Hedgerows have also been modelled to reduce agricultural droughts, as their root systems extend deeper than those of crops or meadows (Table 2). This allows them to access water in deeper and wetter soils layers even though soils in surface were generally drier.

### 3.3.2 Agricultural practices

Soil pitting in maize crop also appeared to enhance modelled infiltration during extreme rainfall events (Figure 6-c), as observed by Clement et al. (2023). However, model outcomes showed a minimal effect of soil pitting on agricultural drought mitigation. This may suggest that extreme rainfall events, when a surplus of infiltration is observed, may be temporally disconnected from droughts events: so, they do not follow close enough together in time for the excess infiltrated water to remain in the root zone until the agricultural drought begins. Another potential factor could be the implementation of hedgerows along field margins in the same scenario, drying soils at the edges and negatively affecting water availability for adjacent crops (Forman and Baudry, 1984).

For reduced tillage practices (Figure 6-b), we did not model large changes in infiltration during extreme events, as reviewed by Clement et al. (2024), nor did we model mitigation of agricultural drought. A well-known positive effect of conservation tillage is the concentration of organic matter into the uppermost soil layer (Haddaway et al., 2017), improving in turn aggregate formation (Chen et al., 2020; Obalum et al., 2019), pore size distribution and soil water retention characteristics (Chandrasekhar et al., 2019). In our model scenario, reduced tillage was virtually implemented by reducing the Van Genuchten alpha parameter of the topsoil by 20 %. This adjustment slightly shifted the soil porosity towards smaller pores, specifically from the range of 0 to -330 hPa to the range of -330 hPa to -15000 hPa, consequently increasing the soil's available water holding capacity. Despite expectations that this improvement in soil physical quality would buffer agricultural drought, our simulation results showed no significant positive effect on mitigating it. Wittwer et al. (2023) recently suggested that while conservation tillage practices could improve soil characteristics, they do not significantly affect

root water uptake patterns under severe drought conditions. Moreover, they emphasized that plant ecophysiology and cropping system diversification in space (crop mixture) and time (crop rotation) (Sanford et al., 2021) have a greater impact on crop performance under drought than soil quality. This is a conclusion that our simulation results seem to support. NbS that were modelled to solely improve the retention characteristics of surface soils without affecting root depth, such as reduced tillage practices or even forest practices aimed at limiting soil compaction, showed a minimal effect on reducing vegetation water stress. One possibility is, during dry periods, soil desiccates from the surface downward over time. Therefore, if the plant root system extends deeper than the surface soil layer, enhancing the retention properties of this superficial layer, where desiccation occurs first, does not seem to be very impactful for increasing plant resilience to prolonged droughts.

### 3.3.3 Forest practices

In the forest context, model results indicate that forest diversification could be effective in improving infiltration during extreme rainfall events and in decreasing days of agricultural droughts (Figure 6-e). Mixed forests are often shown to outperform monocultures regarding agricultural drought resistance, especially in dry areas (Liu et al., 2022; Pardos et al., 2021). In our model scenario, forest diversification was implemented primarily by increasing rooting depth, reflecting the observation that mixed forests often develop deeper and more structurally developed root systems than monocultures, especially spruce even-aged plantations that are common in the C2. This results from the inherent differences in rooting patterns between species, leading to generally deeper root extensions or denser fine root systems in mixed stands compared to pure stands (Jose et al., 2006; Meinen et al., 2009), as well as from the competitive environment in mixed stands, which may encourage greater root investment (Reubens et al., 2007). However, the drought resilience of mixed forests may also be attributed to other mechanisms, such as resource partitioning (e.g., root stratification), facilitation (e.g., active hydraulic redistribution), and selection (e.g., a higher likelihood of including drought-tolerant species) (Pardos et al., 2021). However, these complex mechanisms were not explicitly represented in the model.

Practices aimed at limiting forest soil compaction (Figure 6-d) also appeared to be effective in improving infiltration but exhibited low efficiency in reducing agricultural drought, as rooting depth was not increased with this measure.

In accordance with the current literature (Wastiaux, 2008), our model showed almost no improvement of infiltration and agricultural drought mitigation after peatland restoration in moors or in drained spruce plantations on degraded peat soil (Figure 6-f).

### 3.4 Spatial variability of effectiveness of NbS against flood

This section examines spatial variability and the role of soil characteristics in the effectiveness of modelled NbS against floods, with the aim of identifying the most effective NbS-location combinations. The displayed cumulative infiltration data from 13 to 18 July 2021, before (BASELINE) and after (POST) implementing NbS, are modelled for each natural drainage class (Table 3) from the Belgian soil map (Figure 7). Almost all infiltration distributions exhibited a characteristic bimodal

shape: one peak near zero, where the soils do not or barely infiltrate, and another peak where infiltration approaches the total cumulative rainfall. Based on these distributions, we can identify runoff production zones (below the cumulative precipitation) and runoff interception/reinfiltration zones (above). Grid cells with negative infiltration rates indicate areas where seepage flow is occurring.

Observing the density distributions of infiltration, a shift is evident between non-saturated soils and more saturated soils, with soils without waterlogging generally exhibiting higher infiltration rates than waterlogged soils.

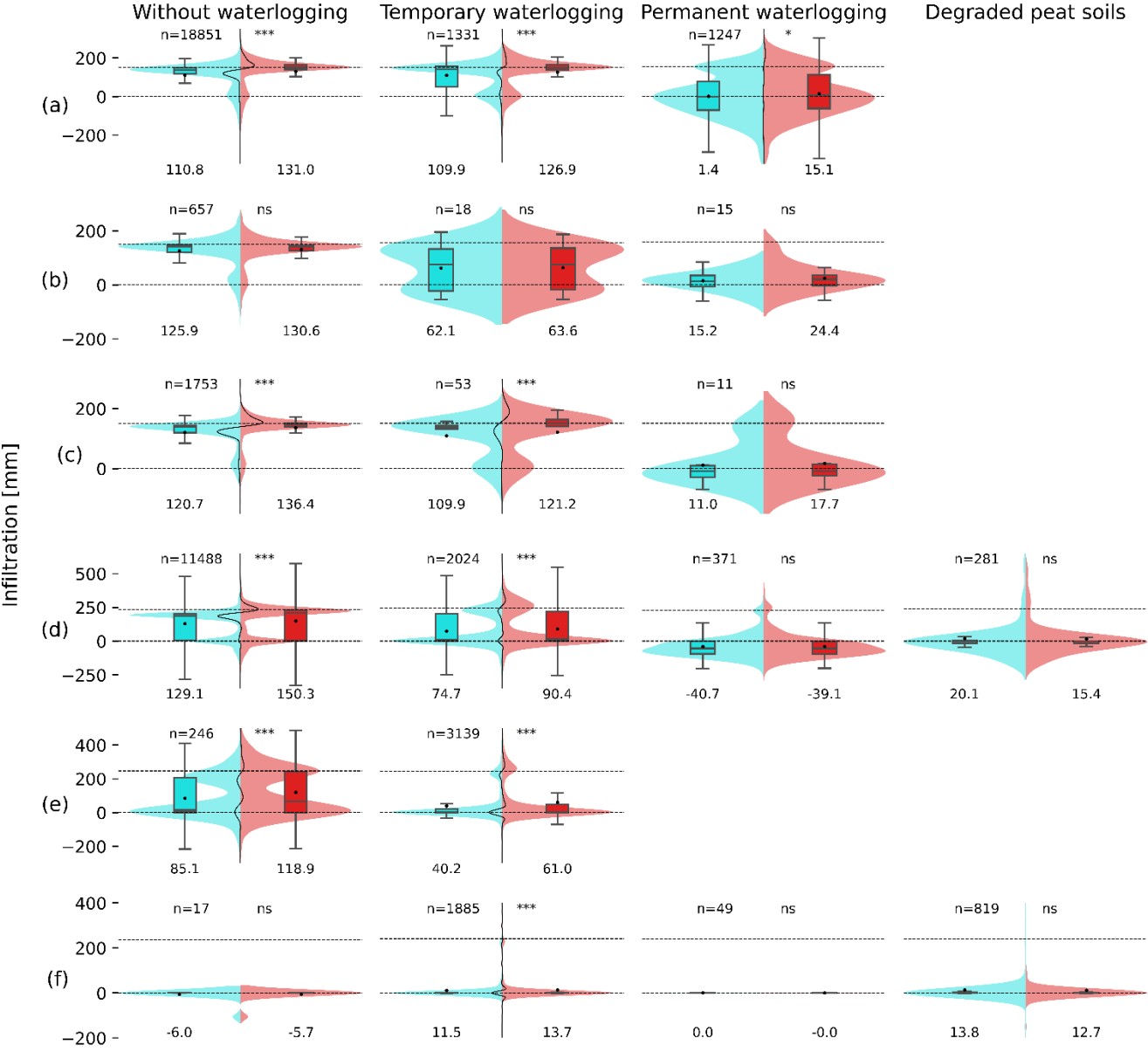

**Figure 7: Comparison of cumulative infiltration (from 13 to 18 July 2021) before (BASELINE) and after (POST) implementing NbS (a : Hedgerows, b : Reduced tillage, c : Soil pitting, d : Forest practices aimed at limiting soil compaction, e : Forest diversification with practices aimed at limiting soil compaction, f : Restoration of peatlands) for each drainage classes from the Belgian soil map. Each data points (n) represent the infiltration of one model grid cell. Significance levels - ns (p-value>0.05), * (0.01<p-value<0.05), ** (0.001<p-value<0.01), *** (p-value<0.001) - indicate results of Kolmogorov-Smirnov tests comparing BASELINE and POST distributions. The centred vertical black lines depict the difference in kernel density estimates between different BASELINE and POST distributions. Numbers below each boxplot and black points are the means. Horizontal dotted lines represent 0 infiltration and mean cumulated rainfall between 13 and 18 July 2021.**

When comparing the BASELINE and POST infiltration distributions and their significance levels, it becomes clear that the effectiveness of NbS in enhancing infiltration diminishes as soil water saturation increases. This trend is observed across a gradient from well-drained soils to temporarily waterlogged areas, degraded peat soils, and permanently waterlogged zones. These findings suggest that well-drained soils may offer greater potential for improving infiltration, implying that, natural soil drainage characteristics should be considered when prioritizing areas of NbS implementation.

This trend might also explain why forest practices aimed at reducing soil compaction have been slightly more effective than forest diversification combined with these practices (Figure 6): the latter were mostly implemented on soils prone to temporary waterlogging, whereas the former were predominantly applied to well-drained soils (Figure 7). In waterlogged soils, the enhanced permeability at the surface did not result in increased infiltration rates, probably because the soil was saturated or close to saturation.

Conversely, in the absence of waterlogging constraints, enhancing soil surface properties seem to significantly increase infiltration. This effect appeared to be further amplified by changes in initial soil moisture conditions before rainfall events, as in the cases of hedgerows and forest diversification (Figure 8). The transition to more evaporative vegetation generally leads to drier surface soils locally. This drying reduces the shallow soil water matric potential, thereby increasing the hydraulic gradient responsible for infiltration. However, the magnitude of this effect on the variation of initial moisture conditions is not uniformly distributed within the watershed, and seems to vary with soils' natural drainage, or averaged saturated zone depth (Figure 8).

In areas where soils are permanently or temporarily waterlogged, the implementation of hedgerows or forest diversification did not affect initial soil moisture conditions or had only a minimal effect. This may be due to the anoxic conditions that might limit root development and transpiration (Caubel, 2001). Additionally, in these areas, increased evapotranspiration may not result in drier soils, as water consistently flows towards these areas from adjacent waterlogged soils.

In initially dry soils, increased evapotranspiration results in somewhat drier soil conditions, but the effect on enhanced infiltration is moderate. The reason for this is that the modelled infiltration capacity of these soils was barely surpassed by rainfall intensity on the extreme event of July 2021 (Figure 8).

In contrast, soils with an average groundwater table depth between 0.5 m and 2 m seem to exhibit optimal conditions for improving infiltration, as evidenced by a significant bell-shaped function (Figure 8). In these areas, increased evaporation enhances unfilled soil macroporosity, also represented by a bell-shaped function (Figure 8). The resulting increased pore space, coupled with a higher hydraulic gradient, may significantly boost infiltration. These soils corresponded to soils without waterlogging with moderate or imperfect drainage, in the Belgian soil classification. Our findings suggest that such areas may be the most favorable for implementing NbS that enhance evapotranspiration, such as hedgerows, to improve infiltration within a watershed.

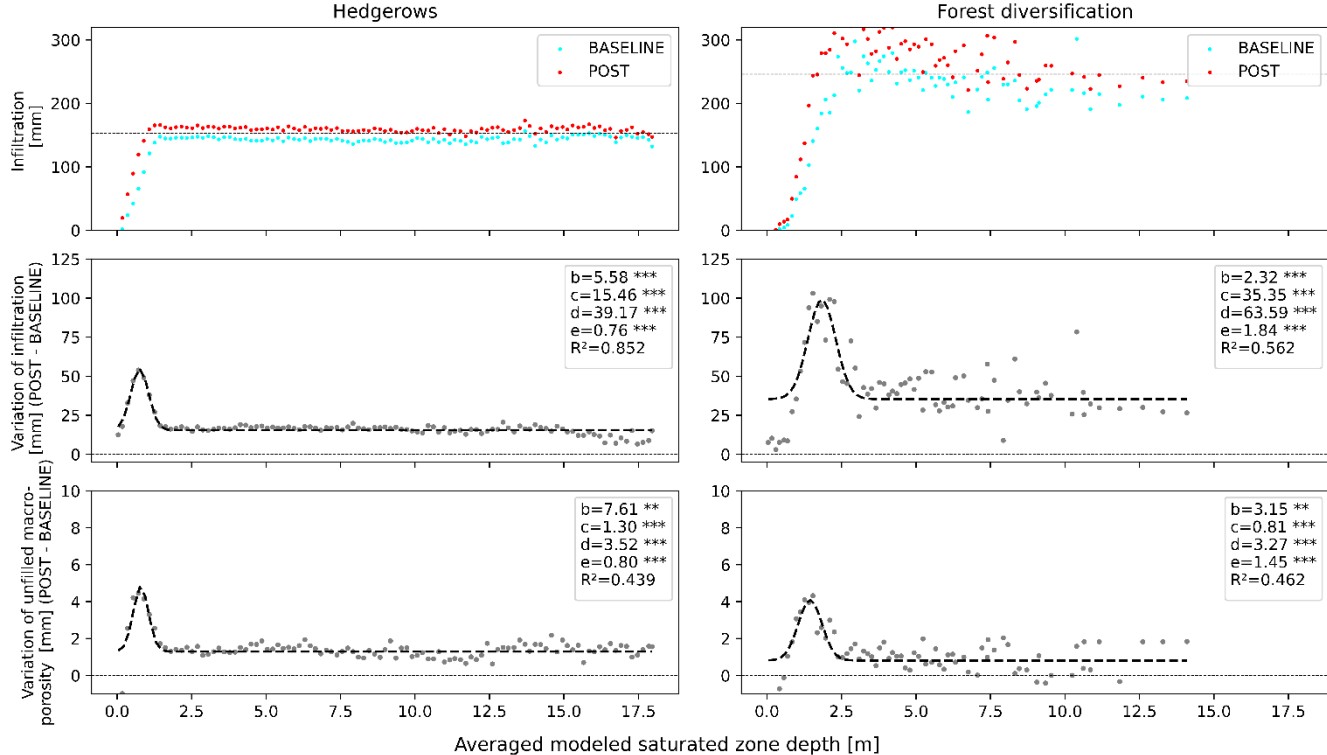

**Figure 8: Impact of hedgerows and forest diversification on infiltration and antecedent soil moisture conditions as function of the average saturated zone depth. Top: Cumulative infiltration from 13 to 18 July 2021. The horizontal line is the spatially averaged cumulated rainfall. Middle: Variation of cumulative infiltration from 13 to 18 July 2021. Bottom: Variation of unfilled macroporosity (porosity defined between 0 and -100 cm of matric potential) in the top 30 cm of soil on 12 July 2021. The abscissa is the average modelled saturated zone depth (from 1 January 2004 to 31 December 2021). Data points represent average observations for fixed saturated zone depth intervals. The dotted line represents a non-linear bell-shaped regression (Bragg : $y = c + (d\,exp^{(-b(x-e)^2)})$) fitted to observed data where : b is the slope around the inflection points of the bell, c is the constant adjusting the minimum, d+c is the maximum value at the top of the bell and e is the x value at which such maximum occurs.**

Additionally, in soils without waterlogging, reinfiltration (also called runon) of runoff generated uphill can occur. This phenomenon is clearly illustrated in Figure 7, where infiltration rates exceed precipitation rates. Typically, runoff-runon processes occur where areas of low infiltration capacity lie upslope of areas with higher infiltration capacity (Woolhiser et al., 1996). Hence, runoff-runon processes highly depend on the spatial heterogeneity of soil infiltration capacity. In our simulations, runoff-runon processes appeared to mostly coincide with areas with good infiltration capacity just downstream of impermeabilized soils, such as roads or urbanized zones. Further research would be valuable in determining whether these areas should be prioritized for implementing NbS aimed at retaining overland runoff and enhancing reinfiltration.

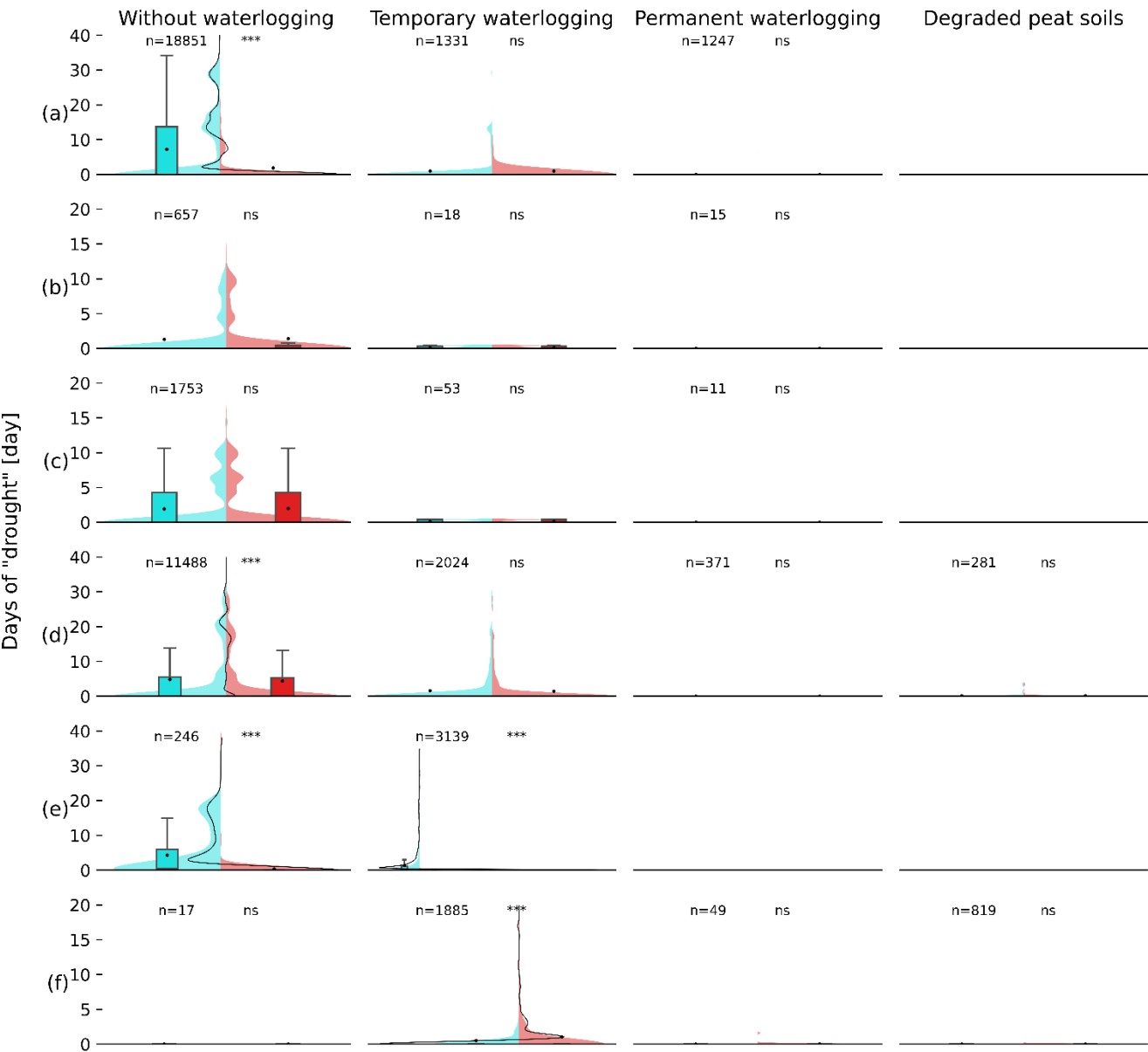

**Figure 9: Comparison of days of drought (matric potential of -30m and return period of 25 years) before (BASELINE) and after (POST) implementing NbS (a : Hedgerows, b : Reduced tillage, c : Soil pitting, d : Forest practices aimed at limiting soil compaction, e : Forest diversification with practices aimed at limiting soil compaction, f : Restoration of peatlands) for each drainage classification from the Belgian soil map. Each data points (n) represent the infiltration of one model grid cell. Significance levels - ns (p-value>0.05), * (0.01<p-value<0.05), ** (0.001<p-value<0.01), *** (p-value<0.001) - indicate results of Kolmogorov-Smirnov tests comparing BASELINE and POST distributions. The centred vertical black lines depict the difference in kernel density estimates between different BASELINE and POST distributions.**

## 3.5 Spatial variability of effectiveness of NbS against agricultural drought

The density distributions of days experiencing vegetation stress across different soil drainage classes reveal that such stresses are most prevalent in soils without waterlogging (Figure 9). The vegetation in temporarily waterlogged areas, degraded peat soils, and permanently waterlogged zones was far less susceptible to these stresses, primarily due to the naturally higher soil water availability in these environments, even after prolonged droughts. Therefore, NbS installed on non-waterlogged/water limited soils were the most effective in limiting agricultural drought. As a concrete example, hedgerows implemented on temporarily waterlogged soils showed a limited effect on drought compared to soils without waterlogging (Figure 9–a).

## 3.6 Synergies and trade-offs between flood and drought mitigation with NbS

The tested NbS primarily had a positive impact on flood reduction with 80.6 % of managed grid cells showing improved infiltration capacity (Figure 10). In contrast, the positive impact on agricultural drought was less significant, with only 27.3 % of managed grid cells exhibiting improvement. However, we observed a minimal negative impact from NbS on agricultural droughts, with only 4.1 % of managed grid cells being affected. In most cases (68.6 % of managed grid cells), there was no discernible impact of NbS on agricultural droughts.

Managed grid cells that were positively affected against both floods and agricultural droughts (25.1 %) were predominantly those with hedgerows and forest diversifications (Figure 10). Nevertheless, we believe that the observed relationship between increased infiltration during extreme rainfall events and reduced vegetation water stress is not causal, but just correlated. This correlation may be primarily explained by changes in vegetation and modelled root depth (Table 2 and Table 4), rather than by an increase in infiltration. Notably, NbS that substantially reduced the number of days experiencing water stress, such as hedgerows or forest diversification combined with practices aimed at limiting soil compaction, also resulted in drier soils due to higher initial interception and evapotranspiration rates. However, a common characteristic of these NbS is the modification of the vegetation and, consequently, root depth, counterbalancing drier upper soil layers. Increased root depth benefits against agricultural drought in two primary ways: it expands the volume of soil reachable to roots, thereby increasing the total available water. Furthermore, deeper root systems enable plants to access deeper and typically wetter soil layers, which can be critical during periods of prolonged drought. It has been reviewed that a deeper root system can significantly enhance drought resistance, particularly in water limited environments where subsoil water is available (Li et al., 2022).

On the other hand, the implementation of NbS that extend root depth can enhance infiltration, as an extensive root network facilitates the formation of macropores within the soil matrix, even after root senescence (Beven and Germann, 1982; Wallace et al., 2021). Additionally, a robust root system supports substantial water uptake to facilitate transpiration, which consequently leads to soil drying (Caubel, 2001). Soil desiccation promotes infiltration by increasing the hydraulic gradient between surface and soil, by creating more air filled spaces for water to infiltrate, and by inducing structural cracks that

facilitate rapid preferential flow (Zhu et al., 2018). However, this very specific phenomenon was not represented in the model.

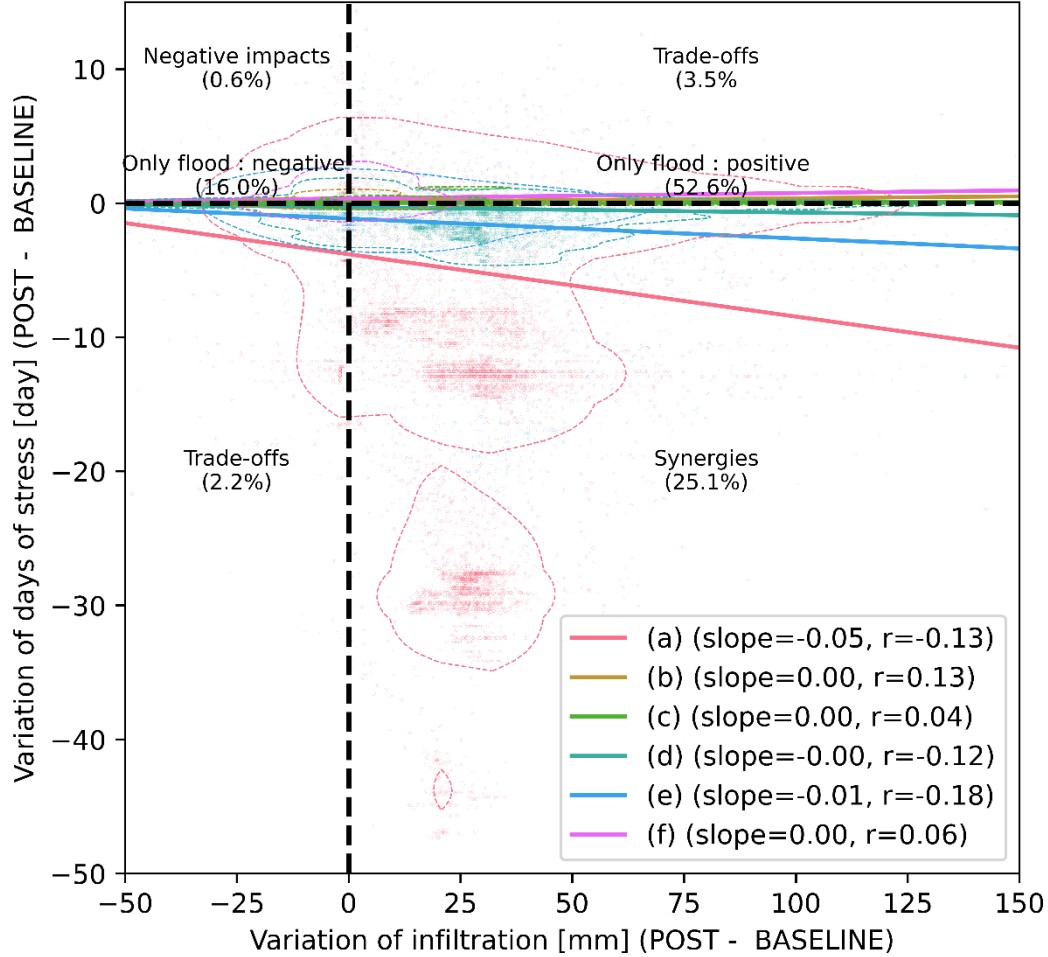

Figure 10: Scatter plot and regression lines of the variation cumulative infiltration (from 13 to 18 July 2021) against days of drought (matric potential of -30m and return period of 25 years ) on model grid cells before (BASELINE) and after (POST) implementing NbS (a : Hedgerows, b : Reduced tillage, c : Soil pitting, d : Forest practices aimed at limiting soil compaction, e : Forest diversification with practices aimed at limiting soil compaction, f : Restoration of peatlands). Vertical and horizontal lines represent thresholds between negative and positive impacts on flood and drought, respectively. Percentages represent the
proportion of total points (model grid cells) inside and between each quadrant.

From this perspective, promoting root growth and sustaining plant transpiration could be considered important to help reduce both vegetation water stress, caused by prolonged drought, and flood risks. In an agricultural context, this can be achieved by limiting soil compaction, which might also hamper infiltration (Unger and Kaspar, 1994). For instance, biopores have been
found to substantially mitigate transpiration deficits during droughts by promoting root systems as preferential growth paths permitting crops to access water from deeper and moister soil layers (Landl et al., 2019). Compaction caused by machinery

traffic in forest soils may also reduce the rooting density of trees due to a lack of soil aeration (von Wilpert and Schäffer, 2006) and an increased ground penetration resistance (Goutal, 2012).

Finally, it is important to note that the balance between synergies and trade-offs is not solely dependent on the type of NbS but also on the location where they are implemented. As discussed in previous sections, NbS were more effective in soils without waterlogging, both in mitigating the risks of flooding and addressing agricultural droughts. As a direction for future work, this opens the door to incorporating allocation change experiments and multi-objective optimisation into our modelling framework. Since different NbS or combinations of NbS can create both benefits and trade-offs across multiple (often conflicting) objectives, it is essential to identify one or several optimal implementation strategies that maximize benefits while minimizing trade-offs. Among others, one approach is based on Pareto-optimality (Deb and Jain, 2014), where a set of optimal solutions is generated such that no objective can be improved without worsening another. From this set of optimal alternatives, decision-makers can explore and select appropriate solutions based on their priorities. Among the available tools for such analyses, the Constrained Multi-Objective Optimization of Land Use Allocation (CoMOLA) software (Strauch et al., 2019) has been developed to facilitate these assessments and has been applied on other NbS modelling frameworks using SWAT+ (European Commission, 2020).

## 3.7 Study limitations and knowledge gaps

### 3.7.1 Data requirements

Representing the hydrological functioning of a watershed at a fine spatial scale remains a challenge for modellers. Physically based distributed hydrological models offer great potential for capturing detailed processes, but they require a large number of parameters to be determined or calibrated. In many cases, the data required for model calibration and validation are often unavailable or highly uncertain at the necessary scale. For instance, calibrating and validating models at the watershed outlet does not necessarily ensure accurate representation of the spatialized hydrological processes within the watershed. Multiple combinations of spatialized model input parameters can yield identical modelled hydrographs at the catchment outlet. To address these challenges, a semi-quantitative spatial validation was carried out by comparing the spatially distributed average depths of the modelled saturated zone with the drainage classes from the Belgian soil map. This exercise demonstrated that the modelling approach effectively captures the fine-scale interactions between soil and water processes. Although this method is specifically tailored to the Belgian context, since this map is specific to Belgium, it provides a valuable framework that could be adapted to other regions. This calls for future research on spatially distributed and cross-scale hydrological observations to further enhance model calibration and validation.

Our results indicated that soil infiltration capacity may be a crucial factor in determining the most suitable locations for implementing NbS. However, accurately representing the spatial and temporal (Pirlot et al., 2024) variability of soil hydrodynamic properties remains challenging with current methodologies. The maps depicting the soils hydrodynamic

properties are inherently limited in their representativeness due to uncertainties in the applied pedotransfer functions, which are primarily derived from agricultural soils, overlooking much of the soil diversity, such as stony soils. Additional limitations arise from uncertainties and the spatial resolution of the underlying soil property maps (e.g., soil depth, texture, organic carbon, and bulk density), as well as the suitability of the continuous function used to describe soil hydrodynamic properties (Weber et al., 2024). Although characterizing the spatio-temporal variability of soil hydrodynamic properties offers valuable benefits in many fields such as hydrology, agriculture, irrigation, ecology or environmental sciences, this remains poorly explored and warrants further research.

Similarly, vegetation plays a crucial role in the soil hydrological functioning at fine scales, significantly influencing watersheds dynamics. A striking example is the impact of planting hedgerows, which locally decreases soil moisture, enhancing infiltration rates, resulting in substantial reductions in peak flow at the watershed outlet. While these effects highlight the importance of vegetation, key aspects of its characteristics (also used as model input), particularly tree rooting systems, remain surprisingly underdeveloped. Despite their critical role, our understanding of roots, including their growth, development, and interaction with complex physical, chemical, and biological soil properties, is still limited (Centenaro et al., 2018).

### 3.7.2 Uncertainties

Three major sources of uncertainty affect estimates of NbS effectiveness in our study: (i) the choice of the hydrological model and its ability to represent key processes, (ii) the model parametrization and modelling of hydrological functioning across the entire catchment in the BASELINE scenario, and (iii) the parameterization of NbS (Brauman et al., 2022). The first source arises from the inherent limitations of any modelling approach, as different models emphasize different processes and may not fully capture key local dynamics. Unlike many other models, the proposed approach explicitly accounts for the spatial variability of local conditions, which may influence NbS effectiveness (Lalonde et al., 2024). However, certain key processes remain unrepresented, for instance, the recycling of atmospheric moisture (Keys et al., 2018; Theeuwen et al., 2023) and the temporal variability of soil hydraulic properties (Pirlot et al., 2024), which may introduce additional uncertainties. Furthermore, it is critical that the model's spatial resolution corresponds to the scale at which NbS are implemented. For example, the simulation of hedgerows is limited by the model's 20-meter resolution, which may not accurately capture their true configuration, introducing uncertainty. The second source is partially addressed through our model validation process, which integrates all available data, including data from gauging stations and drainage classes from the Belgian soil map, enhancing confidence in the model's representation of catchment-scale hydrological dynamics. The third source has not been explicitly quantified. Instead, we compare our results with reported NbS effectiveness in the literature to assess whether our findings align with expected trends. While this provides a useful reference, variations in study contexts may introduce further uncertainties. A sensitivity analysis of NbS parameterization, spanning a plausible

range of parameter values, could offer valuable insights into how parameter input uncertainties influence model output uncertainties. Improving the robustness in the estimation of NbS effectiveness remains a critical area for future research.

### 3.7.3 Multi-scale assessment of NbS

Currently, there is no globally accepted or standardized approach for monitoring NbS (Kumar et al., 2021). The fine-scale spatial variability of effectiveness of NbS demonstrated in our study highlights the need to carefully consider the appropriate methods and scales for monitoring these interventions after implementation. These approaches should be determined based on the type, extent, and expected outcomes of NbS. Macro-scale indicators at the watershed level, such as reductions in peak flow, can be highly useful for assessing the overall, cumulative effects of a combination of NbS implemented upstream. While these indicators are useful, they do not distinguish the individual impacts of specific interventions. Therefore, fine-scale indicators are also necessary to optimize the benefits of NbS. Nevertheless, these assume that individual interventions act independently and do not interact at small scales, an assumption that may not be always true.

Monitoring approaches often rely on ground-based measurement stations, which are often limited in number and scattered sparsely. Encouraging, recent advances in airborne and satellite-based remote sensing technologies may enable the possibility to bridge this gap between scales, improving the capacity to systematically monitor NbS performance (Kumar et al., 2021). For instance, monitoring the effectiveness of NbS in mitigating agricultural drought can be achieved through the comparison of vegetation indices such as the Normalized Difference Vegetation Index (NDVI) or the Vegetation Health Index (Patel et al., 2012). Additionnaly, flood and estimated damage costs might also be evaluated by GIS-based monitoring (Haq et al., 2012).

It is important to recognize that the proposed "Spatial variation of infiltration rates" indicator represents just one aspect of NbS effectiveness in reducing flood risks. While it provides valuable insights into how NbS mitigates the hazard itself, a comprehensive assessment should also account for downstream exposure or vulnerability, which are crucial factors to consider when planning flood risks management (Klijn et al., 2015). For instance, interventions may reduce flood risks locally, but their impacts can also be felt much farther downstream (Lane, 2017). For a comprehensive assessment of the effectiveness of NbS it is crucial to adopt cross-scale approaches since the effects of NbS are not necessarily confined to the immediate area of implementation. Therefore, in-depth evaluation of NbS requires models capable of capturing complex interactions across spatial scales.

Furthermore, the effectiveness of NbS may vary over time, depending on (i) meteorological variability (e.g., antecedent moisture conditions) and (ii) the time required for a measure to become fully effective. For instance, some measures, such as forest diversification, may take several years or even decades to reach full maturity. In the current study, while the effect antecedent moisture conditions was effectively modelled, the transition phase between the BASELINE and POST scenario

was omitted. However, time-variable effects of NbS may be important to consider when defining priorities in a specific local catchment context (Fennell et al., 2023b). These time-variable effects also raises the question of incorporating future climate scenarios into the modelling of NbS effectiveness (Gómez Martín et al., 2021).

### 3.7.4 Multi-objective assessment of NbS

In the present article, the potential synergies or trade-offs that can arise with NbS for mitigating floods and agricultural droughts were evaluated. Beyond these findings, other co-benefits or trade-offs may emerge, which should be assessed according to the specific context and interests of the region in question. For example, afforestation is known to increase watershed evapotranspiration, reducing river flow and groundwater levels. While this trade-off may be limited in humid climates (Hou et al., 2023; Zhang et al., 2017), which are also less prone to hydrological droughts, such practices, like planting alien trees, could be problematic in arid or semi-arid regions that are more sensitive to long-term hydrological and socio-economic droughts (Rebelo et al., 2022). Hence, the impacts of these actions extend beyond hydrological effects, encompassing important social, cultural, economic, and environmental aspects (Sowińska-Świerkosz and García, 2021). NbS are inherently integrative and multifunctional; focusing solely on a hydrological performance without considering the broader social, economic, political (governance) and environmental context can undermine the success and legitimacy of projects (Brauman et al., 2022). This underscores the value of multi-objective optimization procedures, which help identify optimal scenarios that balance competing objectives (Strauch et al., 2019; Castro, 2022; Yang et al., 2023).

### 3.7.5 Bridging the gap between simulated scenarios and real-world implementation

As demonstrated in this study and many others, NbS offer significant potential for mitigating floods and droughts. To translate this potential into real-world implementations, several challenges must be carefully addressed. Raška et al., 2022 identified four key aspects that influence the successful implementation of NbS: i) uncertainties regarding their effectiveness, ii) decisions on optimal locations, iii) institutional frameworks, and iv) resource availability. The first two aspects primarily arise from the inherent variability of NbS effectiveness across spatial and temporal scales, as well as their multi-objective nature - issues that this study, somehow, aims to elucidate. By providing a quantitative assessment of NbS effectiveness, this research contributes to a more informed decision-making process for stakeholders and policymakers, creating a foundation for discussions on implementation strategies. Strong stakeholder engagement plays a key role in ensuring successful NbS deployment. NbS deployment may generate conflicting interests between different groups, creating both winners and losers (Bark et al., 2021). An interdisciplinary approach that integrates hydrology with social sciences such as sociology and economics can help promote transparency and fairness, particularly in designing effective compensation mechanisms that encourage participation. Moreover, aligning NbS implementation with governance structures is essential. While NbS are most effective when they are planned at the watershed scale, decision-making often occurs within administrative boundaries that may not fully reflect hydrological realities. Strengthening coordination between authorities and clearly defining responsibilities will enhance the feasibility of NbS adoption.

## 4 Conclusion

We presented a process-based modelling approach, making the full use of available data, to evaluate the effectiveness of NbS implemented in productive forest and agricultural catchments for flood and agricultural drought mitigation. Rather than focusing solely on the riverbed, the measures considered in this study target catchment slopes, aiming to enhance vertical water fluxes such as infiltration, recharge, and evapotranspiration. These solutions are designed for multifunctionality and are generally easier to implement than hard engineering measures. The modelling approach allowed for the representation of fine-scale interactions between soil and water processes across different spatial scales, which may play a crucial role in the functioning of measures. It also enabled the computation and development of indicators to assess NbS potential for flood and agricultural drought mitigation at multiple scales, focusing on catchment slopes and outlet. Our results emphasized the critical role of spatial variability in NbS effectiveness, influenced by soil characteristics. A key finding is that the effectiveness of NbS in reducing flood risk is significantly influenced by the soil's natural drainage characteristics, which vary across a watershed. In this study, soils with moderate drainage characteristics appeared to show the highest potential for enhanced infiltration through NbS that increase evapotranspiration, such as hedgerows. Additionally, NbS that enhanced root depth were most effective in mitigating agricultural drought, especially in water-limited environments. This study shows that NbS can influence the hydrological functioning of a watershed as a whole, simultaneously affecting both floods and droughts. This underscores the need to move away from siloed evaluations of NbS, but rather adopt multi-scale and multi-objective assessment approaches. The study, acknowledges the challenges of current modelling and monitoring techniques, particularly in the need for measured data at various spatial scales and uncertainty estimates, showing the way for future research. This study encourages decision-makers to envision more hydrologically resilient landscapes - an increasingly urgent necessity in our latitudes.

## Appendix A

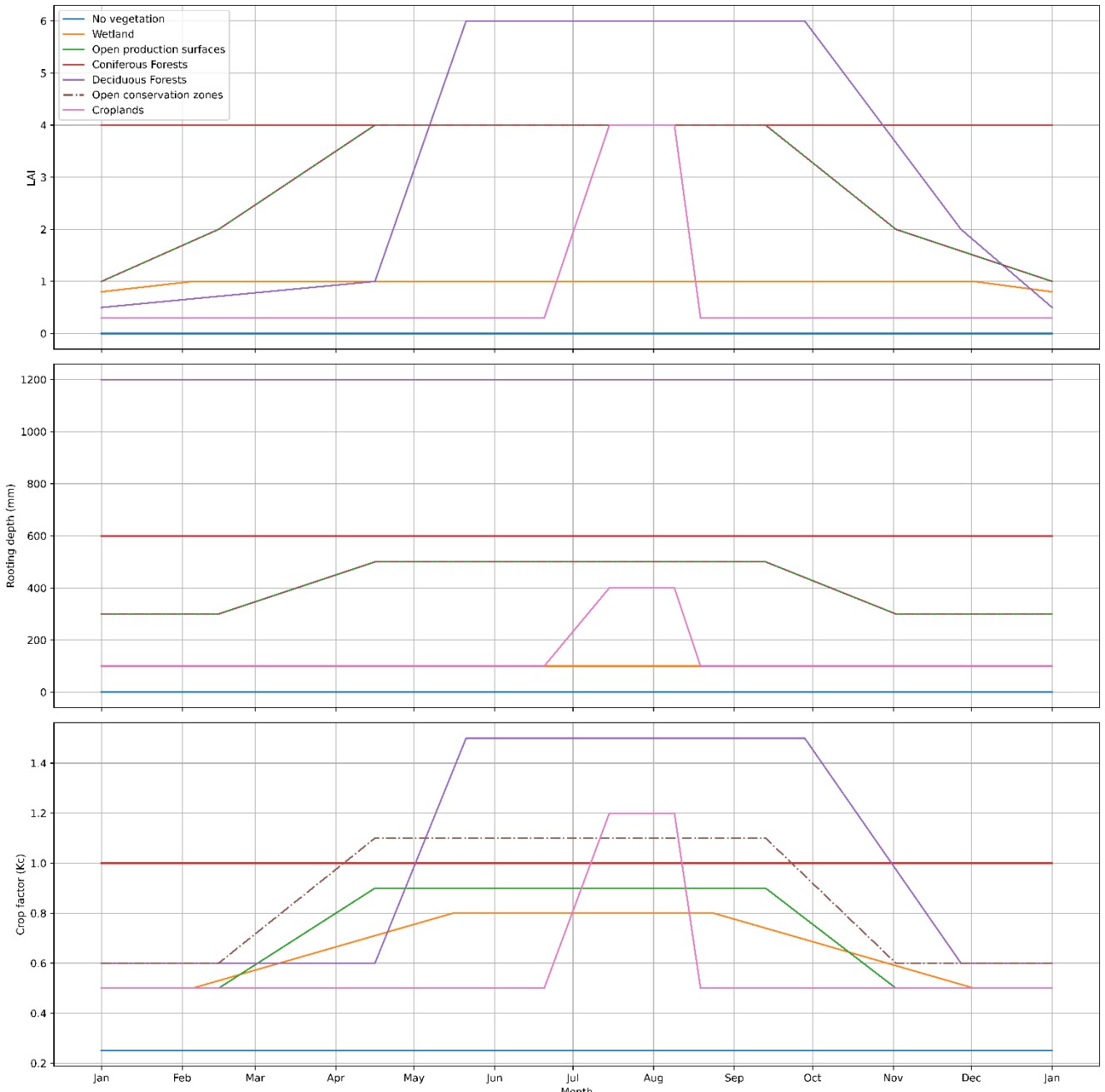

**Figure - A1: Vegetation development trajectories (LAI, rooting depth and Kc) used as model input to calculate evapotranspiration.**

## Code and data availability

Most input data are open source and are cited in the article. MIKE SHE/MIKE 1D is proprietary software. Model output data and codes that support the findings of this study are available upon request.

## Author contributions

BG: Conceptualization; Data curation; Formal analysis; Investigation; Software; Methodology; Visualization; Writing – original draft preparation; Writing – review & editing.

AM: Conceptualization; Data curation; Formal analysis; Investigation; Software; Methodology; Writing – review & editing.

AD: Conceptualization; Methodology; Project administration; Supervision; Funding acquisition; Writing – review & editing.

## Competing interests

The authors declare that they have no conflict of interest.

## Acknowledgements

We thank Noémie Bonaventure, Lisa Di Maggio, Emmanuelle Leyh, and Sara Rabouli for their active participation in the "ModRec Vesdre" project. We also give credit to the Studio023PaolaViganò team (Milan, Bruxelles) and Jacques Teller's team (Uliège) for their contributions to the "Schéma stratégique multidisciplinaire du bassin versant de la Vesdre" (https://territoire.wallonie.be/fr/page/inondations), which was used in this study to develop landscape planning scenarios.

## Financial support

This study was conducted as part of the "ModRec Vesdre" agreement (https://orbi.uliege.be/handle/2268/314437), funded by the SPW Agriculture, ressources naturelles et Environnement  (DGARNE).

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
