# Peer review of "Leveraging soil diversity to mitigate hydrological extremes with nature-based solutions in productive catchments: an application and insights into the way forward"

_EGUsphere, 2024_

## Author Comment (AC2)

**Answer to reviewer 2**

We have added our response below each of your comments (in bold).

**The study highlights the role of nature-based solutions (NbS) in mitigating floods and agricultural droughts, particularly through soil-vegetation approaches that improve soil health and ecosystem resilience. The authors found that local soil characteristics significantly influence NbS effectiveness. Using hydrological models, it showed that well-drained soils enhance flood and drought resilience, as hedgerows improve infiltration and hydraulic properties. In contrast, waterlogged soils show limited benefits due to saturation. The study emphasizes the need to consider spatial soil variability when designing and placing NbS for maximum impact. In general, it is a well-elaborated and well-written manuscript, but I think that the value of the study, its pros, cons and system limits should be pointed out a bit better. I have listed my suggestions below. In times of increasing numbers of floods and droughts, such studies are very relevant in order to be able to derive resilient measures to mitigate the negative effects of these events. For me it is a valuable modelling experiment that cleary shows the potential but also the existing gaps in this field of research.**

**Abstract**

**Well written. I would already here mention what kind of model you used and what NbS you investigated, makes it more specfic. If you cannot mention all, give the most important ones / examples.**

**I think you could also highlight a bit more that is an experimental study (in my opinion), because you also investigate in how far the model is able to simulate the effect of the retention measures. I would write this. Point out what is innovative, what was surprising, and what are pros, cons and limits of your general and modelling approach. It should made be clearer what the value of your study is.**

We agree with your remarks and will revise the abstract accordingly:

- We will specify the type of model used and the NbS investigated.
- Regarding the comment on the 'experimental study,' we will emphasize this aspect more in the abstract. Additionally, we will introduce a specific first objective: "Present a modelling approach that represents both the natural drainage characteristics of the soil and spatialized NbS scenarios, as well as their interactions." We will also review Sections 3.1 and 3.2 in light of this objective.

See a proposed revised abstract :

*Nature-based solutions (NbS) are increasingly being explored as effective strategies for mitigating hydrological extremes, such as floods and agricultural droughts. Among these, soil-vegetation-based approaches may play a key role in improving soil health, enhancing ecosystem services, and restoring the natural hydrological cycle in productive agricultural and forestry catchments, making these landscapes more resilient to climate change. However, the influence of local factors, such as soil characteristics, on the effectiveness of these interventions is often overlooked. This is largely due to the fact that commonly used lumped empirical/conceptual modelling approaches often oversimplify the complex interactions across soil – water processes at multiple scales.*

*This study presents an innovative approach to modelling the effects of NbS landscape planning scenarios, explicitly simulating soil water fluxes. This approach allows for the investigation of how the spatial variability of soil properties influences NbS effectiveness in mitigating both floods and agricultural droughts.*

*A fully distributed, physically based hydrological model was used to represent NbS at the catchment scale, simulating water fluxes (e.g., infiltration, evapotranspiration, runoff, soil water content) on a variable-resolution grid. This model relies on measurable local parameters (e.g., topography, soil properties, vegetation) to capture small-scale hydrological processes, enabling NbS scenarios representation through spatial and temporal parameter adjustments. Hydrological simulations were conducted for two catchments: one agricultural and one forest dominated, each integrating a specific landscape planning scenario. In the agricultural catchment, the scenario involved the integration of hedgerows, reduced tillage practices and soil pitting for maize crops. In the forested catchment, NbS focused on forest diversification, practices aimed at limiting soil compaction in forest and restoration of peatlands. NbS effectiveness was assessed not only downstream but also where they are implemented within the catchment using key spatial indicators. Among them, an indicator evaluating the susceptibility of vegetation cover to water stress at a given location was newly developed.*

*The modelling approach was validated by accurately reproducing river discharge and saturated zone dynamics. It effectively captures soil natural drainage characteristics and provides a reasonable representation of NbS effectiveness, as indicated by consistency between simulated and literature values. A key finding of this study is that NbS effectiveness in enhancing flood and drought resilience strongly depends on soil natural drainage characteristics. In well drained soils, hedgerows significantly enhanced infiltration by improving soil hydraulic properties and creating additional air space in the soil's porosity through higher rates of evapotranspiration. In contrast, improving hydraulic properties in waterlogged soils had minimal impact on infiltration due to saturation, with anoxic conditions potentially limiting transpiration. Additionally, the study highlights that well-drained soils*

*offer co-benefits for resilience to agricultural droughts, as they are more likely to experience water deficits that NbS can mitigate. In contrast, such benefits are generally absent in waterlogged soils, which rarely face water scarcity.*

*Results highlight that the evaluation of NbS effectiveness should recognize the spatial variability in their performance. This variability should inform the type and location of NbS to increase their overall effectiveness. The study underscores the need to move away from siloed evaluations of NbS and instead adopt a coherent, territory-based approach. Our study encourages decision-makers to envision more hydrologically resilient landscapes—an increasingly urgent necessity in our latitudes. This modelling approach provides a solid foundation for future research, with potential for refining NbS effectiveness assessments. Addressing its data requirements and enhancing uncertainty analysis will further strengthen its application.*

**1.Introduction**

**General comments: Well elaborated. For me, however, it is not clear whether the two study areas were already affected by floods and droughts, so perhaps this could be added here (also in the description of the study areas). I hope I did not miss such a statement in the manuscript.**

Yes, the study areas have already been affected by floods and droughts. Initially, we did not plan to expand on the specific characteristics of our study areas in the introduction, as we wanted to emphasize that our study is not just a case study but that our approach could also be valuable for other regions and catchments. Therefore, we only mentioned how floods and droughts affected Central Europe in the last years.

However, we will clarify in the description of the study areas that our specific study area, the Vesdre Valley, was the most affected region in Belgium during the July 2021 floods, with 39 fatalities and material damages amounting to several hundred million euros. Some Belgian climate scientists have modeled that, given climate change, such event could occur again two or three times until 2050, meaning the area remains at risk (https://orbi.uliege.be/handle/2268/312886).

Regarding droughts, we would not say that the study area is more affected than other parts of Belgium. However, Belgium experienced unprecedented droughts in the summers of 2018, 2019, and 2020 (https://orbi.uliege.be/handle/2268/300105). It induced *Picea abies* mortality (https://link.springer.com/article/10.1007/s10661-024-12372-0) and loss of crop productivity (https://www.cra.wallonie.be/uploads/2018/08/S%C3%A9cheresse_2018_en_Wallonie_20 180813-v2.pdf). Climatologists also predict an increase in drought frequency in the coming

decades (especially if global warming exceeds 1.5°C), which could negatively impact the region, especially peatlands (due to a growing summer moisture deficit), *Picea abies* forestry, and agriculture.

**Line 47 ff (NbS): I think that the wealth of different "perspectives" and opinions on NbS can be confusing, you can be a bit more specific regarding your measures (some of them seems to be Natural (Small) Water Retention Measures ;o) ). Have a look at our paper that tries to bring a bit more order into all these concepts:https://www.mdpi.com/2071-1050/16/3/1308. I was thinking of it when reading the concerning sentences in the manuscript. Just have a look at our paper, if it helps, that's fine, if not, that's fine too. No obligation / need to cite, only if it helps.**

Indeed, choosing the right overarching concept to categorize our measures was not straightforward. We had a look at your paper but the question of which term is best suited for our paper remains open, as we believe that each has its pros and cons.

On one hand, NbS might be too broad, as it includes solutions beyond those specifically aimed at water management, which is the main focus of our article. On the other hand, the term Natural (Small) Water Retention Measures (NSWRM) is more specific, as it encompasses only measures primarily intended for water management, even though they may also provide co-benefits such as biodiversity enhancement, climate resilience, etc.

As hydrologists, we might be tempted to classify all our measures as NSWRM, viewing water management as the primary challenge. However, from a forest manager's perspective for example, measures such as forest diversification may serve other primary objectives—forest resilience, biodiversity, aesthetic value, and economic benefits—that are more significant than hydrological regulation.

We will mention other common overarching concepts, including NSWRM, in the introduction and cite your work, which clearly defines these terms. This would give readers a reference for further explanation of these terminologies.

**Line 76 ff (models): You might provide a brief concise overview on some of such models either here or in the model framework description (2.2) which also lead to your model selection.**

Yes, we propose to overview other commonly used models (SWAT+, SCS, HEC-HMS/RAS, etc.) and cite key studies that have used them to assess NbS efficiency. For pros and cons of each model, we would refer directly to 10.1016/j.scitotenv.2021.147058, which provides a comprehensive synthesis.

**For me, it is in general still a problem that the measures we simulate in the model show an effect almost immediately, which does not correspond to reality. Depending on the landscape conditions, there is a delay in these effects (such as retention). If you see this similar, you could discuss that somewhere (intro or discussion, maybe better there).**

Yes we see it similar and, indeed, this aspect deserves to be considered.

The POST scenarios that we modeled were scenarios where measures were considered "mature". We did not model the transition between the BASELINE and POST scenario.

We will clarify this in the 2.2.2 Landscape planning scenario section, and discuss the implication of this hypothesis in the study limitations section.

**Line 86 ff (objectives) For me, as mentioned, it is also an experimental study that tests the capabiities of your (modelling) procedure for your task. I mean you clearly show the pros and cons and gaps (what still has to be done).**

Yes, we will add an objective on that as mentionned above.

**2.Materials and Methods**
**2.1 Study area**

**General comment: See my comment before, add (maybe here) info if the study areas already faced floods and / or droughts..**

**Figure 1: The soil map is a bit hard to read.**

Yes, we will separate the soil map from the figure 1, make it bigger and change the symbology to beter represent the stone content, which is currently hard to read.

**Line 106: Sentence "Apart from peatlands,.." . I think you can delete this sentence, it is already written in line 100 ("Soils are mostly silty.")(except of the stone content) and does not provide additional information).**

Right. In response to Reviewer 1's comment, we planned to rewrite a dedicated section in the Material and Methods on soil description. The soil map (discussed above) will be included in this section.

**2.2 Modelling framework**

**General comment: See my comment in the intro - provide a brief concise description on other models with similar capabilities (or weaknesses) such as MIKE SHE / MIKE 1D.**

**2.2.1 Hydrological model**

**Line 124 ff (data description): I would add a table listing the data and describe the most relevant characterists / information it. See this example (section 2.3 Model inputs) https://www.sciencedirect.com/science/article/pii/S1470160X1931012X?via%3Dihub**

Good idea. We will include the following table :

| Data type | Data source | Resolution | | Descripition |
|---|---|---|---|---|
| | | **Spatial** | **Temporal** | |
| Rainfall | Gridded observational data | 5 km | Daily | Gridded and point observations are combined to obtain hourly 5 km gridded rainfall |
| | https://hydrometrie.wallonie.be | Point observations | Hourly | |
| Reference evapotranspiration | Gridded observational data | 5 km | Daily | Estimated with the Penman-Monteith equation (Allan et al., 1998) |
| River discharge | https://hydrometrie.wallonie.be | Point observations | Hourly | To calibrate and validate the models |
| DEM | LIDAXES (version 2) - MNT, 2023 | 2 m | 2013-2014 | It is a hydrologically conditioned DEM |
| LULC | WALOUS 2018 - Série, 2024; 140 Bassine et al., 2020 | 5 m | 2018 | This data served as model input to describe the spatial distribution of LAI, Kc, Rd and M value of Manning. |
| Soil | « Carte Numérique des Sols de Wallonie » | 1/20.000 | 1947 - 1991 | The soil map of Belgium was used as input to identify homogenious soil unit, stone content and soil depth. It is also used to validate the model with respect to the natural drainage classification. |
| | Textures et fractions granulométriques de référence des sols de Wallonie - Série | 50 m

3 depths (0-40 cm, 40-80 cm, 80-120 cm) | 1947 - 2020 | Map of textural fractions: clay (0–2 µm), silt (2–50 µm), and sand (50–2000 µm). To each layer of each homogenious soil unit is assigned mean values of textural fractions. |
| | Updated European hydraulic pedotransfer functions (euptfv2) | / | / | Retention and hydraulic conductivity curves each layer of each homogenious soil unit were derived from the pedotransfer function 1 using depth and mean values of textural fractions as predictors |
| Geology | Sohier, 2011 | 1 km | / | This data originate from a Bibliographic study of the hydrogeological context of the Vesdre catchment. It gives the spatial repartition of homogenious hydrogeologic units and an initial guess of hydraulic conductivities, which was refined during the modle calibration. |

**Line 137 ff: What about land management (tillage crop rotation, fertilization, etc.) data, for instance from agricultural statistics, data from Integrated Administration and Control System (IACS) or interviews?**

To our knowledge, systematic statistics on agricultural practices (tillage or fertilization, ...) are almost non-existent or not publicly available in our study areas. In Wallonia, we have a public database that is anonymized and updated annually based on farmers' declarations, providing information only on the main crop of the current year. In our study areas, apart from permanent grasslands, most main crops were maize, as shown in this figure illustrating crop rotation dynamics in the Vesdre catchment. (https://orbi.uliege.be/handle/2268/296474). Apologies, the figure is only available in French.

[Figure]

The LULC map we used incorporates data from this database (2018), which enables us to distinguish different types of crops. Therefore, we decided to simplify croplands by considering two categories: Open production surfaces and croplands (see table 1), assuming maize as a representative crop for parameterizing vegetation development (LAI, Kc and root depth) in crops.

**Line 201 (Moriasi et al., 2007): I know Daniel Moriasi and I highly appreciate his work, and I know that many people use this performance guideline, but I think it is not always the best method to evaluate model performance. In some cases it might not say very much**

**about how well the hydrological dynamics are represented by the model (you describe the dynamics later in the text, so all fine (just a comment)).**

We agree with your point. The Moriasi guidelines are widely used, and we included them as a reference to provide readers with a familiar benchmark. However, we acknowledge their limitations and discuss them in the text (line 534 - 540).

**2.2.2 Landscape planning scenarios**
**General comment: Are these scenarios and suggestions just your ideas or are they based on some River basin or landscape management plans or something like that in your region that suggest some of them?**
**Would be good to know.**

The measures we tested are based on a strategic territorial development framework called the *Schéma stratégique multidisciplinaire du bassin versant de la Vesdre* (Strategic Multidisciplinary Plan for the Vesdre River Basin). This study was conducted in response to the July 2021 floods and aims to rethink territorial development in the Vesdre basin in all its aspects, including hydrography, agriculture, forestry, natural areas, public spaces, mobility, and urbanization. The overarching goal is to make the region less vulnerable to future floods and droughts.

Rather than providing a concrete action plan, the study outlines a long-term vision for sustainable territorial development, proposing a range of potential measures. Our role was to assess the potential effectiveness of such measures proposed in this vision that may have an effect on the catchment's hydrology at mid-long term using hydrological modeling. To do so, we tested three scenarios, each implementing as many feasible measures as possible within a specific context (agriculture, forests, and peatlands), to estimate their maximum potential for flood and drought mitigation. A report has been produced :

https://orbi.uliege.be/handle/2268/314437

This was briefly mentioned in the acknowledgements, but we will cite the Strategic Multidisciplinary Plan for the Vesdre River Basin directly in the 2.2.2 Landscape planning scenarios section, and elabore about (like explained above).

**Line 229 ff (Agricutural practices): These are partly very small areas (if I understand this correctly). Is it relevant?**

Yes, these crop areas cover only 2.6% of the total catchment 1 surface. Consequently, their impact on the hydrograph at the catchment scale (or outlet) is probably limited due to their small extent.

We should, however, mention that the agricultural practice we implemented (soil pitting for maize crops) is an incremental rather than a transformational measure. Unlike no-till farming, it does not require farmers to change their agricultural system.

Our scenarios were aimed at implementing as many feasible measures as possible in our study site to demonstrate their potential combined maximal effecteveness while staying true to the catchment's existing conditions (LULC). We sought to demonstrate a range of options, which we also evaluate directly where they are implemented. This is one reason we have chosen a fully distributed hydrological model: it allows us to assess the effectiveness of measures not only at the outlet but also within the watershed, directly where they are implemented and effective.

Is it relevant? We believe that when planning flood risk reduction through hydrological measures, "*every square meter counts*".

**Table 3: The table is titled "summary of hypotheses.." - are there other papers that used these or similar parameter modifications to "describe" such measures? How to confirm or reject the hypotheses? By measurements? These are these experiments? I guess there is no kind of a parameter database for such measures in the model (similar to SWAT) that can be extended?**

Yes, these are indeed hypotheses, and we acknowledge that some values were chosen somewhat arbitrarily and sometimes even based on "expert judgment". MIKE SHE does not have a database of widely accepted parameter values for these measures, unlike SWAT model with the "Conservation Practice Modeling Guide for SWAT and APEX" from Waidler et al., 2011.

Some of the values we used can be found in the literature. For example, to modify the saturated hydraulic conductivity (Ksat) of the soil beneath hedgerows, we based our approach on the experimental results of Holden et al. (https://www.sciencedirect.com/science/article/pii/S0167880918304894. Other values are derived from measurements conducted in our laboratory, such as the assessment of roughness at the base of hedgerows using hydraulic resistance measurements (https://intelleau.wixsite.com/projet), and also found in literature (https://theses.fr/2018AMIE0031). In certain cases, parameterization follows a logical approach. For instance, when modifying the vegetation type, parameters such as LAI, crop coefficient (Kc), and root depth are adjusted based on pre-existing values in the BASELINE model (see Appendix 1).

Although reference values from previous studies could be used, we cannot assume they apply directly to our specific case. Claiming otherwise would overlook the uncertainty

associated with these values. Therefore, we consider them as hypotheses, and our results depend on these assumptions.

We believe that a key advantage of our modelling framework is its reliance on theoretically measurable parameters that represent a physical reality. In contrast, more empirical and conceptual model-based approaches modify and calibrate "conceptual" parameters—such as the CN value (used in SWAT?) to represent a measure's effect on infiltration—based on expected impacts rather than direct measurements. This advantage could be enhanced through synergies with post-implementation monitoring, as parameter values could be refined using direct measurements after implementation, further improving the credibility and reliability of model outputs. Finally, a sensitivity analysis of final results concerning the choice of these values (covering a plausible range) would add significant value to our approach. However, it is story that, we believe, extend beyond the scope our paper. Nevertheless, this could lead to potential future research.

This limitation of our approach and the associated knowledge gap will be discussed further in the "3.7 Study limitations and gaps" section.

**Is your model able to simulate the effect of each of these measures in equal quality? Or are some "better simulated" than others?**

We believe that the model adequately represents the key hydrological processes affected by these measures, such as infiltration, evapotranspiration, and runoff. However, as with any model, it remains a simplification of reality, meaning that certain processes are not explicitly accounted for. For example, the formation of cracks during soil desiccation, which can lead to preferential flow pathways (line 508), the recycling of atmospheric moisture (line 574) are not included or the temporal variability of soil hydraulic propeties (line 542 : https://doi.org/10.1111/ejss.13558)

As for whether the model simulates the effect of each measure with equal accuracy, we are unfortunately unable to provide an answer at this stage. To do so, we would need direct field measurements of key hydrological variables—such as infiltration rates, evaporation, and soil moisture content—at locations where these measures have been implemented. Such data would allow for a direct comparison with model outputs and provide concrete validation of the model's ability to represent these measures accurately.

**2.2.3 Integrated hydrological analysis of model outputs**

**Line 248 (Hydrological indicators): So, you developed these indicators? I think there several indicators out like this, I think you should provide an overview. Infiltration and**

**agricultural drought, etc., are not new "indicators". I would better point out what is new here (because - again - you write that you developed them) and what is better compared to the exiting ones. Do you see it as as a kind of "enhancement" to other hydrological indicators, such as the "Indicators for Hydrological Alteration (IHA)", introduced mainly by Richter et al. and Poff et al. and many more?**

Yes, the names we assigned to the indicators we calculated or developed may not have been well chosen. We will rename them. We also will clarify which indicators were simply calculated (already existing) and which were newly developed.

The indicators we referred to as "Outlet hydrograph" and "Infiltration" can be found in the literature. For the first indicator ("Outlet hydrograph"), we will rename it as "peak flow percent reduction" and "flood volume percent reduction", which are commonly found in literature. For the second indicator ("Infiltration"), a more precise term would be "spatial variation of infiltration rates." While this type of indicator is not new, to our knowledge, it is rarely used to assess the effectiveness of NBS from a spatially explicit perspective at the subplot scale on slopes. This aspect, we believe, adds some originality to our study.

The last indicator, which we called "Agricultural drought," is more novel (we developed it), as we have not seen it in other studies. It simultaneously considers soil water status (characterized by soil matric potential at different depths) and the vegetation's ability to extract water (characterized by rooting depth). This indicator reflects the susceptibility of the vegetation cover to water stress at a given location, characterized by duration as a function of stress intensity (defined by different matric potential thresholds) and return periods. A more appropiate name for this indicator would be "spatial Vegetation Drought Stress Duration Frequency (sVDSDF)".

Our study aims not only to assess hydrological impacts at the river scale (on river flow regime) but also to consider the entire watershed, including slopes and other relevant processes such as infiltration or susceptibility of the vegetation cover to water stress. We selected and developed these indicators to effectively represent the impact of NbS measures (even if we could be more exaustive) while ensuring they are easily interpretable by decision-makers, who are often non-specialists. This aspect, we believe, differentiate our approach from the "Indicators for Hydrological Alteration (IHA)," which were primarily designed to assess hydrological impacts on aquatic, wetland and riparian ecosystems with an analisys of the river flow regimes, with less emphasis on catchments slopes/drainages areas.

**3.Results and discussion**

**General comment: Well elaborated and written.**

**3.4 Spatial variability of effectiveness of NbS against flood**

**Line 426 ff (These findings..). The question is again) if your results - considering all uncertainties - justify such a statement (second part of the sentence)?**

Maybe we should adopt more cautious terms regarding this sentence. We plan to reformulate the sentence with :

*"These findings suggest that well-drained soils may offer greater potential for improving infiltration, implying that, natural soil drainage characteristics should be considered when prioritizing areas of NbS implementation."*

**What about the delay of measures effects?**

See our response above.

**Regarding flood protection measures - when are floods are too strong, that measures have no effect anymore?**

This paragraph concerns results of effectiveness of measures in term amount of infiltrated water focussing on the rainfall event between 13 and 18 July 2021 which was the most severe ever recorded in the region, with an estimated return period of 300–400 years. These results are therefore relevant to extreme flood events. But yes, with climate change such an event, or even more severe ones, occurring again cannot be ruled out.

**Line 468 ff ("These areas, we believe.." ff): See my comments above. Are the results reliable enough to state this - would you tell that a stakeholder / water manager, etc.? OK, you eaken your statement with last sentence, but think of reformulate it.**

Yes. We propose to rephrase by :

*"Further research would be valuable in determining whether these areas should be prioritized for implementing NbS aimed at retaining overland runoff and enhancing reinfiltration."*

**3.5 Spatial variability of effectiveness of NbS against agricultural drought**

**General comment: You could perform allocation change experiments" (searching for the most effective measures (or measure combinations) at specific locations)? We use multi-criteria optimization for that (see https://www.optain.eu; see also project deliverables)). Could be a point for discussion?**

**3.6 Synergies and trade-offs between flood and drought mitigation with NbS**

**General comment: See my previous comment regarding "allocation experiments" / combination of measures.**

That is something we had not considered, mainly because we relied on the "*Schéma stratégique multidisciplinaire du bassin versant de la Vesdre*" (Strategic Multidisciplinary Plan for the Vesdre River Basin) which served as a reference for our scenarios. But yes, we had a look at your "D5.1: Common optimisation protocol" from the OPTAIN project and your paper on CoMOLA (https://doi.org/10.1016/j.envsoft.2019.05.003). We found your approach very interesting and it deserves to be discussed in our article.

If we understand correctly, a major limitation to applying this approach to our particular modelling framework is the number of simulations/scenarios required to explore the solution space and converge towards a set of optimal solutions. This would require a significant simplification of our models in order to reduce computation times. We believe that SWAT+ would be a better tool to perform such multi-criteria optimization procedure. We propose, though, to extend our last paragraph in section 3.6, mentioning your work:

*"Finally, it is important to note that the balance between synergies and trade-offs depends not only on the type of NbS but also on their spatial implementation. As discussed in previous sections, NbS were more effective in soils without waterlogging, both in mitigating flood risks and addressing agricultural droughts. This opens the door to allocation change experiments and multi-objective optimization.*

*Since different NbS or combinations of NbS can create both benefits and trade-offs across multiple (often conflicting) objectives, it is essential to identify one or several optimal implementation strategies that maximize benefits while minimizing trade-offs. Among other, one approach is based on Pareto-optimality (Deb, 2014), where a set of optimal solutions is generated such that no objective can be improved without worsening another. From this set of optimal alternatives, decision-makers can explore and the select appropriate solutions based on their priorities.*

*Among the available tools for such analyses, the Constrained Multi-Objective Optimization of Land Use Allocation (CoMOLA) software (Strauch et al., 2019) has been developed to facilitate these assessments and has been applied, notably, within the OPTAIN project ([www.optain.eu](http://www.optain.eu))."*

**3.7 Study limitations and knowledge gaps**

**General comments: In general, I would emphasize the value of the study a little better.**

Indeed, in this section, we are—as we believe rightly so—as critical as possible of our approach. We aim to be transparent by acknowledging that our approach is not perfect and

that many questions remain unanswered. However, we believe, this does not diminish the value of our study in any way.

From our perspective, the value of our study lies in its process-based approach, making full use of the available data. We demonstrate that our modeling approach effectively captures the fine-scale interactions between soil and water processes across multiple spatial scales. This enables us to assess the effectiveness of measures in detail, considering the interactions occurring at different scales.

The measures we consider are "nice to have" solutions, offering multifunctionallity and being easier to implement than hard engineering measures. Rather than focusing solely on the riverbed, they target catchment slopes, aiming to boost vertical water fluxes such as infiltration, recharge, and evapotranspiration. At the same time, we evaluate their effects on both floods and droughts.

This underscores the need to move away from siloed evaluations of NbS, which are still common among land managers whose governance structures remain highly compartmentalized. Instead, a coherent, territory approach is essential. Our study encourages decision-makers to envision more hydrologically resilient landscapes—an increasingly urgent necessity in our latitudes.

We will rework on this section with this, to emphazise a bit more the value of the study. We will also addapt our conclusion and abstract accordingly.

**So far, the gaps are highlighted quite well in this chapter (although the word "however" is used a little too often ;o) ), but it should be worked out a little better. What about the uncertainties (capability of the model(s) (what are the system limits (scale, describing the efficincy measure under the given circumstances, data, etc.? Can they (uncetainties) be quantified? What does that do to the reliability of the results? Is it sufficient so far to be able to derive reliable recommendations?**

Yes we will rework this section following your recommandations.

Model uncertainties:  We used all available data, including all gauging stations in the study area and drainage classes from the Belgian soil map, to validate our model results as thoroughly as possible. However, this was likely not sufficient to ensure that we accurately captured the hydrological functioning across the entire study area. We acknowledge this limitation and discuss it in the first paragraph of Section 3.7: Study Limitations and Knowledge Gaps. Additionally, we conducted tests on the uncertainty introduced by pedotransfer functions (using uncertainty estimates from EU-PTF V2) by simulating hydrographs with an alternative model in a smaller catchment. Such analyses, we believe,

extend well beyond the scope of this article, and we plan to address them separately in a future study.

Model capability : As previously discussed, the model remains a simplification of reality, omitting certain subtle processes—such as the temporal variability of soil hydrodynamic properties. However, we believe that it adequately represents key hydrological processes influenced by measures, including infiltration, evapotranspiration, and runoff.

Can uncertainties be quantified ? : Partially, through our model validation procedure. Regarding the representation of measures, our model results align with expectations based on existing literature on NbS effectiveness. However, as noted above, quantifying uncertainties would require direct field measurements of key hydrological variables—such as infiltration rates, evaporation, and soil moisture content—at locations where these measures have been implemented. A sensitivity analysis of measure parameterization could provide valuable insights but would require a dedicated, comprehensive study.

What does that do to the reliability of the results?  Is it sufficient so far to be able to derive reliable recommendations? :

As you suggested in lines 426 and 468, we will review the manuscript to take a more cautious approach when drawing recommendations from model outputs with unverifiable uncertainty. Nevertheless, since we used a process-based modeling approach, even if the exact efficiency values remain uncertain, we can still identify which processes are affected by which measures. This allows us to formulate logically sound recommendations. Our key message in this article remains that local soil drainage characteristics should be considered when implementing  NbS as it influence their effectiveness.

**For me, this is a very good modeling experiment that could be used as a basis for further discussion (scientists, stakeholders), also to specify what needs to be tackled urgently next. To do this, you could involve farmers, authorities, economists, and first ask them for their opinion on the methods, measures and results.**

Yes, we actually used this study to convice to stakeholders from the Vesdre Valley that these measures could be effective. This study enabled us to secure a new project aimed at creating "laboratory" catchments of around 2 km$^2$ in the Vesdre Valley, where such measures would be implemented in practice. The first phase involves iteratively developing a coordinated management plan with various stakeholders in the catchment—authorities, farmers, businesses, and local residents. As hydrologists, our role is to propose an "optimized hydrological management plan", including a quantified assessment of its effectiveness. This plan would then be reviewed by stakeholders to ensure its feasibility and acceptability. Iteratively, we would model the revised plan to assess its hydrological effectiveness.

Ultimately, we hope that such a plan will be implemented, giving us the opportunity to monitor changes in the catchment's hydrological functioning, evaluate the effectiveness of the measures in practice, the changes on local hydrological parameters and addapt the models (parameter values, etc) as function of.

In this section, we will emphasize that while such a (our) study, although essential to demonstrate the potential effectiveness of these measures in order to convince, is not sufficient to guarantee their concrete implementation. To move forward, a collaborative territorial approach with all stakeholders is needed, one that also integrates human sciences.

**I have already mentioned other points (taking into account the delayed effect of the implemented measures, experiments on the spatial allocation of measures (and the combination of measures), flood protection measures – when are floods too strong to be mitigated by these measures? etc.). You basically already have many of the points in this chapter, but I would discuss it a bit more clearly and in a more structured way and – as I said – emphasize the value of the study more clearly.**

See our responses above.

---

## Author Response (AR1)

**Point-by-point reply to the comments**

**Reviewer 1**

- **The manuscript is engaging, well-written, and provides valuable insights into the effectiveness of Nature-Based Solutions (NBS) in the two selected case studies. However, I believe the authors should further clarify the differences in soil characteristics between the two case studies. This is critical since a key focus of the manuscript is on how soil properties influence the efficiency of NBS. Although the two case studies differ in geomorphologic and land use/land cover properties, they share similar soil characteristics (both predominantly silty). Emphasizing this aspect would help differentiate the results and strengthen the analysis of case-specific outcomes.**

Our intent in the manuscript was not to contrast the effectiveness of nature-based solutions (NbS) between the two catchments but rather to compare the effectiveness of different NbS (related to land use) across different pedological contexts (primarily based on natural drainage characteristics) that are present in both catchments, albeit in different proportions. However, we agree that a more detailed description of the soils in the study area would be beneficial.

To address this, we have added a dedicated section titled **Soil Context** in the **Materials and Methods (Study Area)** section, providing a more comprehensive description of the soil units. In this section, we describe the key geological and pedological processes that have shaped the current soil characteristics in the study sites. We also explain how the soils differ, with a particular focus on their natural drainage properties and spatial distribution.

- **Additionally, I suggest some improvements to the figures, as the distinction between the two case studies is not particularly clear in some of them (e.g., Figure 3 and Figure 5). Enhancing the visual representation could better convey the comparative analysis.**

For figure 3, we have added a stacked bar chart of the proportion of surface of natural drainage classes of soils found in the two case studies. This would help to differentiate the catchment in terms of soil natural drainage characteristics.

For figure 5, we have splitted the figure in two parts; one for each case study.

- **Lastly, lines 312–315 are particularly intriguing. The unexpected results warrant a deeper explanation. It would be beneficial if the authors elaborated on the**

**potential reasons behind these findings, focusing on how geomorphological and soil characteristics of the selected areas may have influenced the outcomes. Given the proximity of the two areas, climatic factors are likely not a significant contributor, so discussing the impact of local geomorphology and soil properties in greater detail would add valuable context.**

To avoid any ambiguity, we must clarify that lines 312–315 refer to the peak discharges modelled for the BASELINE scenario (without NBS implementation). This clarification have been be added to the manuscript.

The paragraph primarily aims to recontextualize the two catchments by explaining that, in the current (baseline) scenario, the forest-dominated catchment (C2) generates more runoff than the agriculturally dominated catchment (C1). In the manuscript, we use the term "counterintuitive" to describe the observation that the C2 generates more runoff than the agriculturally dominated catchment C1. However, "counterintuitive" might be too strong a term, as these results are entirely expected and align with our understanding of hydrological functioning of both catchments. Observations at the monitoring stations further confirm this, so it is not surprising. Our intention was simply to highlight that this does not align with the common assumption that forest-dominated catchments produce less runoff than agriculturally dominated ones.

We have addepted the paragraph in order to clarify the interpretation.

Also we have replace the term "climate" with "precipitation rate", which is more valid.

We have also added a sentence about the impact of local soil properties on the modeled hydrological functioning of the catchments, referring to the "soil context" section in the Materials and Methods, and the geological and soil maps figure.

Since this paragraph is intended solely to recontextualize the two catchments by referring exclusively to the BASELINE scenario—and given that the primary objective of the article is not to compare their hydrological functioning under this scenario but rather to assess the effectiveness of NBS—we prefer to keep it concise and maintain focus on the article's main message.

**Reviewer 2**

- **Abstract**
  **Well written. I would already here mention what kind of model you used and what**

**NbS you investigated, makes it more specfic. If you cannot mention all, give the most important ones / examples.**

We have added a small description of the model we used and also mentionned the investigated NbS.

- **I think you could also highlight a bit more that is an experimental study (in my opinion), because you also investigate in how far the model is able to simulate the effect of the retention measures. I would write this.**

We highlighted this point by mentionning that : "This study presents an innovative approach to modelling the effects of NbS landscape planning scenarios, explicitly simulating soil water fluxes. This approach allows for the investigation of how the spatial variability of soil properties influences NbS effectiveness in mitigating both floods and agricultural droughts."

And that : "The modelling approach was validated by accurately reproducing river discharge and saturated zone dynamics. It effectively captures soil natural drainage characteristics and provides a reasonable representation of NbS effectiveness, as indicated by consistency between simulated and literature values."

- **Point out what is innovative, what was surprising, and what are pros, cons and limits of your general and modelling approach. It should made be clearer what the value of your study is.**

We have modified the end of the abstract, covering these aspects : "Results highlight that the evaluation of NbS effectiveness should recognize the spatial variability in their performance. This variability should inform the type and location of NbS to increase their overall effectiveness. The study underscores the need to move away from siloed evaluations of NbS and instead adopt a coherent, territory-based approach. Our study may serve as a basis for discussion and action, supporting decision-makers in implementing measures to enhance hydrological resilience. This modelling approach provides a solid foundation with potential lines of futur research to refine NbS effectiveness assessments, such as strengthening data availability and working on uncertainty analysis."

- **1.Introduction**

  **General comments: Well elaborated. For me, however, it is not clear whether the two study areas were already affected by floods and droughts, so perhaps this could be added here (also in the description of the study areas). I hope I did not miss such a statement in the manuscript.**

We have added a paragraph on this in in the first part of the study area description, focussing on what was the motivation to choose this area specifically.

- **Line 47 ff (NbS): I think that the wealth of different "perspectives" and opinions on NbS can be confusing, you can be a bit more specific regarding your measures (some of them seems to be Natural (Small) Water Retention Measures ;o) ). Have a look at our paper that tries to bring a bit more order into all these concepts:https://www.mdpi.com/2071-1050/16/3/1308. I was thinking of it when reading the concerning sentences in the manuscript. Just have a look at our paper, if it helps, that's fine, if not, that's fine too. No obligation / need to cite, only if it helps.**

We have added a mention on other overarching concept, including NSWRM, in the introduction and we have cited your work, which clearly defines these terms. This would give readers a reference for further explanation of these terminologies.

- **Line 76 ff (models): You might provide a brief concise overview on some of such models either here or in the model framework description (2.2) which also lead to your model selection.**

Yes, we have added this paragraph in the introduction to adress you comment: "Several hydrological models have been used to simulate the effectiveness of NbS, including, among others, SWAT+ (European Commission, 2020), LISFLOOD (Institute for Environment and Sustainability : Joint Research Centre et al., 2012), HEC-HMS (Guido et al., 2023), or MIKE SHE (Fennell et al., 2023b). These models generally integrate different modules to calculate different water fluxes (infiltration, runoff, evapotranspiration) and can be based on empirical, conceptual, or physically-based approaches. For example, SWAT and HEC-HMS use the well-known Soil Conservation Service (SCS) method to calculate water infiltration into the soil based an empirical formula and the soil curve number (CN). Other models, such as MIKE SHE, are process-based and use physical equations, such as Richard's equation, to model infiltration. Some models allow for fully distributed simulations, while others are lumped models or adopt a non-distributed approach. Each model has its own advantages and limitations, and the choice depends on the specific needs of the study in terms of modeling capabilities, as well as the required spatial and temporal scale and resolution, and the available data (Kumar et al., 2021b)."

- **For me, it is in general still a problem that the measures we simulate in the model show an effect almost immediately, which does not correspond to reality. Depending on the landscape conditions, there is a delay in these effects (such as**

**retention). If you see this similar, you could discuss that somewhere (intro or discussion, maybe better there).**

Yes we see it similar, even though the main focus of the artcile is more on the spacial variability of NbS effectiveness, the temporal variability deserves also to be considered.

To do so, we have added in the section "**2.2.2 Landscape planning scenarios**" :

"The scenarios thus compare the current situation with a long-term projection (horizon 2050), without representing the transition process. It should be emphasized that accounting for the transition phase may be essential, as some measures require several years or even decades to achieve full efficiency, and the speed at which a measure becomes effective may serve as an important criterion for prioritizing NbS."

We also discuss this aspect in the "**3.7 Study limitations and knowledge gaps**" section :

"Furthermore, the effectiveness of NbS may vary over time, depending on (i) meteorological variability (e.g., antecedent moisture conditions) and (ii) the time required for a measure to become fully effective. For instance, some measures, such as forest diversification, may take several years or even decades to reach full maturity. In the current study, the transition phase between the BASELINE and POST scenario was omitted. However, time-variable effects of NbS may be important to consider when defining priorities in a specific local catchment context (Fennell et al., 2023b). This time-variable effects also raises the question of incorporating future climate scenarios into the modeling of NbS effectiveness (Gómez Martín et al., 2021)."

- **Line 86 ff (objectives) For me, as mentioned, it is also an experimental study that tests the capabiities of your (modelling) procedure for your task. I mean you clearly show the pros and cons and gaps (what still has to be done).**

Yes, we have added an objective on that :

"The paper addresses four key objectives: i) Building a modelling approach that represents both the spatial variation of soil charateristics (focussing on soil natural drainage) and spatialized NbS scenarios, as well as their interactions..."

And we have added in results that :

"Overall, the model successfully represents the variability in the natural drainage characteristics of soils in the BASELINE scenario. This addressed our first objective, which was a prerequisite for assessing the effectiveness of the NbS measures on various soils with differing natural drainage characteristics."

- **2.Materials and Methods**
  **2.1 Study area**
  **General comment: See my comment before, add (maybe here) info if the study areas already faced floods and / or droughts.**

See above.

- **Figure 1: The soil map is a bit hard to read.**

Yes, we have separated the soil map from the figure 1, to make it bigger and we have changed the symbology to make it clearer.

- **Line 106: Sentence "Apart from peatlands,.." . I think you can delete this sentence, it is already written in line 100 ("Soils are mostly silty.")(except of the stone content) and does not provide additional information).**

This has been removed. And we have rewrited a dedicated section in the Material and Methods on soil description.

- **2.2 Modelling framework**

  **General comment: See my comment in the intro - provide a brief concise description on other models with similar capabilities (or weaknesses) such as MIKE SHE / MIKE 1D.**

We have added a paragraph in the introduction that provides an overview of other hydrological models used to simulate NbS. To avoid excessive repetition, we do not repeat this information here. But only focus on MIKE SHE / Mike 1D capabilities, and how it is addapted to our task.

- **2.2.1 Hydrological model**

  **Line 124 ff (data description): I would add a table listing the data and describe the most relevant characterists / information it. See this example (section 2.3 Model inputs)https://www.sciencedirect.com/science/article/pii/S1470160X1931012X ?via%3Dihub**

We have included a table that summarizes input data.

- **Line 137 ff: What about land management (tillage crop rotation, fertilization, etc.) data, for instance from agricultural statistics, data from Integrated Administration and Control System (IACS) or interviews?**

To our knowledge, systematic statistics on agricultural practices (tillage or fertilization, …) are almost non-existent or not publicly available in our study areas. In Wallonia, we have a public database that is anonymized and updated annually based on farmers' declarations, providing information only on the main crop of the current year. In our study areas, apart from permanent grasslands, most main crops were maize. The LULC map we used incorporates data from this database (2018), which enables us to distinguish different types of crops. Therefore, we decided to simplify croplands by considering two categories: Open production surfaces and croplands assuming maize as a representative crop for parameterizing vegetation development (LAI, Kc and root depth) in crops.

In the article, we have added : "Apart from permanent grasslands, which were considered open production surfaces, most croplands were cultivated with maize. Consequently, maize development dynamics were assigned to all croplands, and in the absence of information on cropping practices, no crop rotations were modelled."

- **Line 201 (Moriasi et al., 2007): I know Daniel Moriasi and I highly appreciate his work, and I know that many people use this performance guideline, but I think it is not always the best method to evaluate model performance. In some cases it might not say very much about how well the hydrological dynamics are represented by the model (you describe the dynamics later in the text, so all fine (just a comment)).**

We agree with your point. The Moriasi guidelines are widely used, and we included them as a reference to provide readers with a familiar benchmark. However, we already acknowledge their limitations and discuss them in the text (line 534 – 540 on the first preprint).

- **2.2.2 Landscape planning scenarios**
  **General comment: Are these scenarios and suggestions just your ideas or are they based on some River basin or landscape management plans or something like that in your region that suggest some of them?**
  **Would be good to know.**

We have added in the beginning of the section  "Landscape planning scenarios":

"These landscape planning POST scenarios were developed in accordance with the "Schéma Stratégique Vesdre" (Inondations - Reconstruction, 2024), which outlines a long-term vision for sustainable territorial development and proposes a range of potential measures to mitigate flood and drought risks. The modelled scenarios (POST) incorporate as

many feasible measures as possible within specific contexts (agriculture-dominated and forest-dominated) to assess their potential for flood and drought mitigation."

The "Schéma Stratégique Vesdre" is also presented at the beging of the presentation of the study area.

- **Line 229 ff (Agricutural practices): These are partly very small areas (if I understand this correctly). Is it relevant?**

Yes, these crop areas cover only 2.6% of the total catchment 1 surface. Consequently, their impact on the hydrograph at the catchment scale (or outlet) is probably limited due to their small extent.

We should, however, mention that the agricultural practice we implemented (soil pitting for maize crops) is an incremental rather than a transformational measure. Unlike no-till farming, it does not require farmers to change their agricultural system.

Our scenarios were aimed at implementing as many feasible measures as possible in our study site to demonstrate their potential combined maximal effecteveness while staying true to the catchment's existing conditions (LULC). We sought to demonstrate a range of options, which we also evaluate directly where they are implemented. This is one reason we have chosen a fully distributed hydrological model: it allows us to assess the effectiveness of measures not only at the outlet but also within the watershed, directly where they are implemented and effective.

We did not make any changes to that point.

- **Table 3: The table is titled "summary of hypotheses.." - are there other papers that used these or similar parameter modifications to "describe" such measures? How to confirm or reject the hypotheses? By measurements? These are these experiments?**
  **I guess there is no kind of a parameter database for such measures in the model (similar to SWAT) that can be extended?**

We have added a disclaimer statemeent about this limitation of our approach in the material and methods :

"These NbS measures are incorporated into the model by adjusting some key parameter values (**Error! Reference source not found.**). It should be noted that although some literature exists on how NbS may influence these parameters, significant uncertainties remain regarding their exact values, as they cannot be assumed to apply directly to this specific case. These parameter values should be considered as hypotheses, while remaining within plausible ranges."

We also discuss this limitation in the "3.7 Study limitations and gaps / Uncertainties" section
:

"Three major sources of uncertainty affect estimates of NbS effectiveness in our study" ...
"(iii) the parameterization of NbS" ... "The third source has not been explicitly quantified.
Instead, we compare our results with reported NbS effectiveness in the literature to assess
whether our findings align with expected trends. While this provides a useful reference,
variations in study contexts may introduce further uncertainties. A sensitivity analysis of NbS
parameterization, spanning a plausible range of values, could offer valuable insights into
how parameter input uncertainties influence model output uncertainties. Improving the
robustness in the estimation of NbS effectiveness remains a critical area for future research."

- **2.2.3 Integrated hydrological analysis of model outputs**
  **Line 248 (Hydrological indicators): So, you developed these indicators? I think
  there several indicators out like this, I think you should provide an overview.
  Infiltration and agricultural drought, etc., are not new "indicators". I would better
  point out what is new here (because - again - you write that you developed them)
  and what is better compared to the exiting ones. Do you see it as as a kind of
  "enhancement" to other hydrological indicators, such as the "Indicators for
  Hydrological Alteration (IHA)", introduced mainly by Richter et al. and Poff et al.
  and many more?**

Yes, the names we assigned to the indicators we calculated or developed may not have been
well chosen. We have renames them. We also have clarified which indicators were simply
calculated (already existing) and which were newly developed.

- **3.Results and discussion**

  **General comment: Well elaborated and written.**
  **3.4 Spatial variability of effectiveness of NbS against flood**

  **Line 426 ff (These findings..). The question is again) if your results - considering
  all uncertainties - justify such a statement (second part of the sentence)?**

We have addapted this sentence. We also reviewed the manuscript to formulate more
nuanced recommendations given all model uncertainties.

- **What about the delay of measures effects?**

See above.

- **Regarding flood protection measures - when are floods are too strong, that measures have no effect anymore?**

This paragraph concerns results of effectiveness of measures in term amount of infiltrated water focussing on the rainfall event between 13 and 18 July 2021 which was the most severe ever recorded in the region, with an estimated return period of 300–400 years. These results are therefore relevant to extreme flood events.

- **Line 468 ff ("These areas, we believe.." ff): See my comments above. Are the results reliable enough to state this - would you tell that a stakeholder / water manager, etc.? OK, you eaken your statement with last sentence, but think of reformulate it.**

We have addapted this sentence.

- **3.5 Spatial variability of effectiveness of NbS against agricultural drought General comment: You could perform allocation change experiments" (searching for the most effective measures (or measure combinations) at specific locations)? We use multi-criteria optimization for that (see https://www.optain.eu; see also project deliverables)). Could be a point for discussion?**
  **3.6 Synergies and trade-offs between flood and drought mitigation with NbS General comment: See my previous comment regarding "allocation experiments" / combination of measures.**

We have extended our last paragraph in section 3.6, discussing multi-criteria optimization :

"This opens the door to allocation change experiments and multi-objective optimization. Since different NbS or combinations of NbS can create both benefits and trade-offs across multiple (often conflicting) objectives, it is essential to identify one or several optimal implementation strategies that maximize benefits while minimizing trade-offs. Among other, one approach is based on Pareto-optimality (Deb and Jain, 2014), where a set of optimal solutions is generated such that no objective can be improved without worsening another. From this set of optimal alternatives, decision-makers can explore and the select appropriate solutions based on their priorities. Among the available tools for such analyses, the Constrained Multi-Objective Optimization of Land Use Allocation (CoMOLA) software (Strauch et al., 2019) has been developed to facilitate these assessments and has been applied, notably, within the OPTAIN project (European Commission, 2020)."

- **3.7 Study limitations and knowledge gaps**

  **General comments: In general, I would emphasize the value of the study a little better.**
  **So far, the gaps are highlighted quite well in this chapter (although the word "however" is used a little too often ;o) ), but it should be worked out a little better. What about the uncertainties (capability of the model(s) (what are the system limits (scale, describing the efficincy measure under the given circumstances, data, etc.? Can they (uncetainties) be quantified? What does that do to the reliability of the results?  Is it sufficient so far to be able to derive reliable recommendations?**

We have completely revised Section 3.7 to make it more structured and readable. We have organised it into five subsections:

- Data requirements

- Uncertainties

- Multi-scale assessment of NbS

- Multi-objective assessment of NbS

- Bridging the gap between simulated scenarios and real-world implementation

The comments you made regarding uncertainties are mainly addressed in the "Uncertainties" sub-section.

We have also rewritten this 3.7 section to make it more positive, highlighting through various examples the value of our study, outlining what remains to be done, and suggesting future research directions.

- **For me, this is a very good modeling experiment that could be used as a basis for further discussion (scientists, stakeholders), also to specify what needs to be tackled urgently next. To do this, you could involve farmers, authorities, economists, and first ask them for their opinion on the methods, measures and results.**

We tried to address this comment by adding a new subsection devoted to "**3.7.5 Bridging the gap between simulated scenarios and real-world implementation**".

- **I have already mentioned other points (taking into account the delayed effect of the implemented measures, experiments on the spatial allocation of measures (and the combination of measures), flood protection measures – when are floods**

**too strong to be mitigated by these measures? etc.). You basically already have many of the points in this chapter, but I would discuss it a bit more clearly and in a more structured way and – as I said – emphasize the value of the study more clearly.**

See our responses above.

---

## Author Response (AR2)

**Point-by-point reply to the comments**

**Anonymous referee #3**

- **This paper presents a comprehensive modeling study examining the effectiveness of nature-based solutions (NbS) for mitigating both floods and agricultural droughts in Belgian catchments. The authors use a physically-based hydrological model (MIKE SHE/MIKE 1D) to evaluate various NbS measures and emphasize the importance of soil drainage characteristics in determining their effectiveness. While the study addresses an important and timely topic with a reasonable methodological approach, several significant concerns limit the strength of the conclusions. The main issues relate to model validation, oversimplified parameter representations, and methodological limitations in the comparative analysis between catchments. Despite these limitations, the work provides valuable insights into spatial targeting of NbS and contributes to the growing literature on nature-based flood and drought management. Detailed comments are provided below, which I hope will be useful in clarifying and strengthening the manuscript:**

Thank you for your review. Below, we provide answers to all your comments and explain how we have modified the manuscript accordingly.

- **Page 13, Line 295: I'm concerned about how the authors validated their saturated zone modeling. They mention doing a "visual assessment" against soil drainage classes (Table 3), but don't explain what this actually means or how they did it. Without any real groundwater measurements to compare against, they're basically checking if their model matches the same soil data they used to build it in the first place - which isn't very convincing validation.**
  **Since the whole paper hinges on how soil drainage affects these nature-based solutions, this validation is concerning. The authors should either explain their validation approach much better or be more upfront about this major limitation.**

We acknowledge that this validation approach is experimental and still open to improvement. However, we still think it is valuable as it offers the advantage of leveraging the limited data available in the study site to assess whether the simulated hydrological behavior of soils reasonably reflects field observations, and does so in a fully spatially distributed way. This remains uncommon in hydrological modelling, where validation often relies on a

relatively sparsed point-based measurements (gauging stations, piezometers, sometimes soils moisture probes).

We would also like to clarify that the soil drainage class information from the Belgian soil map is mostly independent of the parameterization of soils in the model. It is used only for the spatial discretisation of soil units. The hydraulic properties assigned to each soil unit, including retention and conductivity curves, are not derived from the drainage classes. Instead, they are generated solely using the EUPTFv2 pedotransfer function, based on an independent map of soil textural fractions and the depth of the soil horizon as predictors.

Moreover, the modelled dynamics of the saturated zone are influenced not only by the soil pararmetrization but also by other input parameters such as topography, geology, vegetation and climate. This validation therefore helps to verify that the combination of these parameters results in a reasonably coherent representation of soil functioning across the model domain.

In the manuscript, we have revised line 295 to clarify this approach and have added a direct reference to Figure 4, which presents the outcome of the validation.

- **Page 15, Table 4: The translation of nature-based solutions into model parameters seems overly simplistic. For instance, representing soil pitting as just adding 20mm of water storage ignores the actual geometry and connectivity of these features. Similarly, hedgerows create complex root systems and flow pathways that go well beyond modifying just the top 40cm of soil properties. The authors should elaborate on this.**

Yes, we agree with your point. It was also raised by the reviewer 2 in his initial review. You mentioned two examples, but there are likely many more; at this stage, evaluating the appropriateness of NbS parameterization remains largely based on expert judgement. However, we believe the advantage of our approach to parametrize NbS, compared to more conceptual modelling strategies, is that it is grounded in physical parameters, which are, for the most part, measurable. As the literature on NbS continues to grow, this opens the door to improved parameterization based on concrete field measurements rather than on calibration of model parameters. However given the current state of knowledge, we prefer consider our NbS parametrization as hypotheses (we recognized the uncertainty associated), and our results depend on these assumptions.

This is why we mentioned just above the table 4 ( in lines 336 – 340 ) that : *"It should be noted that although some literature exists on how NbS may influence these parameters, significant uncertainties remain regarding their exact values, as they cannot be assumed to apply*

*directly to this specific case. These parameter values should be considered as hypotheses, while remaining within plausible ranges.”*

We also acknowledged this limitation in our section 3.7.2 (line 680) : *“Three major sources of uncertainty affect estimates of NbS effectiveness in our study: … (iii) the parameterization of NbS (Brauman et al., 2022) … The third source has not been explicitly quantified. Instead, we compare our results with reported NbS effectiveness in the literature to assess whether our findings align with expected trends. While this provides a useful reference, variations in study contexts may introduce further uncertainties. A sensitivity analysis of NbS parameterization, spanning a plausible range of parameter values, could offer valuable insights into how parameter input uncertainties influence model output uncertainties. Improving the robustness in the estimation of NbS effectiveness remains a critical area for future research.”*

We have added a few words to this study limitations section emphasizing the need for more field-based evaluations of NbS effectiveness, under various conditions, which would permit direct comparisons with model outputs and provide validation of the model's ability to represent these measures accurately.

- **Page 19, Lines 415-419: The comparison between the two catchments has some fundamental issues that weaken the conclusions about soil drainage effects. The authors are comparing completely different systems - different geology, topography, precipitation rates (C2 gets 20% more rainfall), and entirely different types of NbS measures (hedgerows vs. peatland restoration). With so many variables changing at once, attributing the differences in effectiveness solely to soil drainage characteristics is problematic. A fair comparison could be testing the same measures across areas with different drainage within the same catchment.**

In this section, our intention was not to attribute the differences in overall scenario effectiveness to the natural soil drainage characteristics of the catchments. The purpose was simply to present the results of the two scenarios, which were implemented in different catchments, and developed in accordance with the predominant land use (agricultural or forested) by de "Schéma stratégique Vesdre". This section should be seen as a preliminary result intended to contextualise the rest of the results. It was not our aim to explain the origin of these differences here. As you rightly pointed out, this would require a more in-depth analysis.

We have added a sentence after line 433 stating that these results compare the overall outcomes of two different scenarios applied in two distinct catchments, which differ in land

use, geology, topography, and precipitation rates. Therefore, these results should not be used to generalize the effectiveness of the specific measures implemented within each scenario.

- **Page 25, Figure 5: The spatial effectiveness analysis presents interesting findings, but there are some methodological considerations that could strengthen the conclusions. The study uses the Belgian soil drainage classification for both model validation and to demonstrate that soil drainage controls NbS effectiveness, which creates some circularity in the reasoning.**

We do not see a fundamental issue in using the Belgian soil drainage classification both for model validation and to illustrate that soil drainage influences NbS effectiveness, as the validation was carried out solely on the "baseline" scenario and the Belgian soil drainage classification was not used to calibrate any of the NbS measures.

- **The statistical analysis in Figure 7 would benefit from addressing the highly unbalanced sample sizes across drainage classes (n=15 to n=18,851), which may affect the reliability of the significance tests.**

Yes, this difference in sample size between drainage classes is explained by the uneven distribution of land use types across these classes. For example, in agricultural contexts, most croplands are established on well-drained soils, while waterlogged soils are generally avoided. This naturally leads to highly unbalanced sample sizes when analysing the effects of NBS by drainage class.

We adapted the significance tests by applying them to a subset of 200 data points randomly sampled. Significance tests results were removed for samples smaller than 200 data points. We also have corrected Figure 9 in the same way. Some significance tests results slightly changed but in general our previous observations and our reasoning holds, so we didn't made any change in the main text.

- **Additionally, since different NbS measures were implemented in different locations, it's challenging to isolate whether effectiveness differences stem from soil drainage characteristics or from the spatial distribution of the measures themselves.**

Yes, we agree with this point. At the very least, there appears to be a correlation between the improvement of infiltration and the soil drainage class where some measures are implemented. But is this truly due to the drainage class itself, or to other correlated factors,

such as topography? It is clear that other factors also play a role, and we have clarified this point more explicitly in the manuscript.

That said, we are strongly convinced that this is not merely a correlation, but that there is indeed a causal link between soil drainage class and the effect of certain measures. This is what we aim to explore further in the following discussion (lines 540–560) and in Figure 8, by analysing some of the mechanisms that could explain this causal relationship.

- **The analysis focuses on one extreme event (July 2021), and it would be valuable to see how these patterns hold across different hydrological conditions. The authors acknowledge that their 20-40m model resolution may limit representation of fine-scale processes, which is particularly relevant for practical NbS placement decisions.**

This figure presents the same analysis as Figure 7, but for a different rainfall event (from 21 to 23 August 2007) with a total precipitation of approximately 50 mm (compared to approximately 200 - 250 mm for July 2021). Significance tests were also performed on a sample of 200 points. In our view, the observed patterns remain relatively consistent. To maintain coherence and avoid further lengthening an already dense manuscript, we prefer to leave this information here and not include it in the main text.

[Figure]

- **Page 30, Figure 10: The synergies and trade-offs analysis is interesting but feels somewhat predetermined by the study design. Since the authors specifically chose measures that enhance root systems (hedgerows, forest diversification) and placed them primarily in well-drained soils, it's not surprising that these same measures show benefits for both floods and droughts. The conclusion that root depth is the key mechanism linking both benefits makes sense conceptually, but the analysis doesn't really test this hypothesis. A more convincing analysis would compare the same measures across different soil types or test measures that specifically don't involve root system changes.**

We are aware that our approach is only one among others, and that it can be improved. In the manuscript, we already acknowledge the limitations of our approach, including

uncertainties, and highlight that it opens the door to new perspectives and research directions.

Here (bellow) is the comparison you suggested: a figure similar to Figure 10, but focusing only on hedgerows (panel a, left), which involve changes in root depth in the modelled scenario, and forest practices aimed at limiting soil compaction (panel d, right), which do not involve changes to the rooting system. Both are tested across different natural soil drainage classes. These additional results appear to support our conclusions.

While we find this additional result interesting and consistent with our main findings, we feel it does not substantially enhance the value of the manuscript and primarily serves as further support for conclusions already well established in the main text. Given the manuscript is already quite dense, we prefer not to include it in the main body. We still remain open to including it as an annex if the editorial team deems it necessary.

[Figure]

- **The low percentage of areas showing drought benefits (27.3%) also raises questions about whether these measures are really as effective for drought mitigation as suggested.**

This low percentage is due to the fact that measures were not specifically implemented in what we defined as drought-prone areas, as it could be observed from the figure 9 on the distributions of days of drought (matric potential of -30m and return period of 25 years) before (BASELINE) implementing NBS. As a result, their potential to mitigate drought could not be fully expressed in locations that were not affected by drought conditions.

**Referee #2: Martin Volk**

- **Dear authors,**
  **you have done a good job and the quality of the manuscript has improved considerably. You have considered and discussed all my suggestions and comments, and in an excellent way. The only thing that needs to be changed is the abstract. At over 550 words, it is far too long. It needs to be shortened. And keywords need to be added.**
  **Thank you and best wishes,**
  **Martin Volk**

Thank you, Mr Martin Volk, for you review. We have shortened the abstract to approximately 320 words. Unfortunately, we could not find any specific guidelines regarding keywords in the HESS submission instructions, and did not find where to place them. If possible, we suggest to include the following keywords : Nature-based solutions, Floods, Agricultural drought, Hydrological modelling, Soil properties.

---

## Author Response (AR3)

**Author's Response**

We have adapted the colour schemes of Figs. 3, 10, and A1 to make them more accessible to readers with colour vision deficiencies.